# FAITHFULFACES: POSE-FAITHFUL FACIAL IDENTITY PRESERVATION FOR TEXT-TO-VIDEO GENERATION

## ABSTRACT

Identity-preserving text-to-video generation (IPT2V) empowers users to produce diverse and imaginative videos with consistent human facial identity. Although existing open-source and commercial methods have demonstrated impressive performance in typical scenarios, they still face significant limitations when confronted with challenging cases, such as large facial pose variations or facial occlusions. These challenges frequently result in identity distortion in the generated videos. In this paper, we propose *FaithfulFaces*, a pose-faithful facial identity preservation learning framework to improve IPT2V in complex dynamic scenes. Specifically, FaithfulFaces first proposes a pose-shared identity aligner that refines and aligns facial poses across distinct views via a pose-shared dictionary and a pose variation–identity invariance constraint. Then, the well-learned aligner can capture the global facial pose representation from the input single-view face image with explicit Euler angle embeddings, which could provide a pose-faithful facial prior for foundational generative models to better preserve identity in the generated videos. In particular, we develop a high-quality video dataset pipeline featuring substantial facial pose variations specifically for our FaithfulFaces to facilitate robust training. Compared to other IPT2V methods, FaithfulFaces achieves state-of-the-art performance across multiple metrics, generating high-quality videos with clear facial structures and consistent identity preservation, even as facial pose changes and occlusions occur. The code and dataset pipeline will be released.

## 1 INTRODUCTION

Identity-preserving text-to-video generation (IPT2V) is a specialized facet of content creation that aims to generate various videos from the user-provided reference image and text prompts while maintaining consistent human facial identity across consecutive frames Yuan et al. (2025); Xue et al. (2025). This task showcases the potential to create and author visual content across domains, including but not limited to film and television production, personalized avatars, advertising design, and social multimedia content.

Benefiting from the robust generative capabilities of the large-scale pre-trained video foundational generative models Kong et al. (2024); Yang et al. (2025); Wan et al. (2025), the IPT2V task can seamlessly extend these models to generate videos guided by reference face images. To generate videos with high-fidelity facial identity, researchers have proposed various methods to represent the identity information of the reference image. For example, ID-Animator He et al. (2024) used a lightweight face adapter to encode the identity-relevant embeddings. ConsisID Yuan et al. (2025) designed two facial extractors to extract global low-frequency structure and local high-frequency details for IPT2V. At the same time, many commercial video generation models, such as Vidu Vidu (2025), Kling Kling (2025), have also been adapted to the IPT2V task. Therefore, this task is the focus of the generative AI field and has attracted widespread attention.

Despite their notable success, existing methods still exhibit limitations in effectively handling certain intricate scenarios. As shown in Fig. 1, we visualize the generation results of different methods in a complex dynamic case, where ConsisID and VACE Jiang et al. (2025) are two representative open-source methods, based on CogVideoX-5B Yang et al. (2025) and Wan2.1-14B Wan et al. (2025), respectively. Kling is one of the most popular and powerful commercial models. In this case, the goal is to generate a video depicting a subject performing a boxing action, which often involves

Input Reference Image and Text Prompt

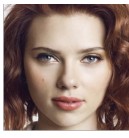

*The video features a person in a gym setting, wearing a light gray long-sleeve shirt and black boxing gloves. The individual is engaged in a boxing routine, demonstrating various punches and defensive maneuvers...*

Open-Source: ConsisID

Open-Source: VACE

Commercial: Kling

Our Method

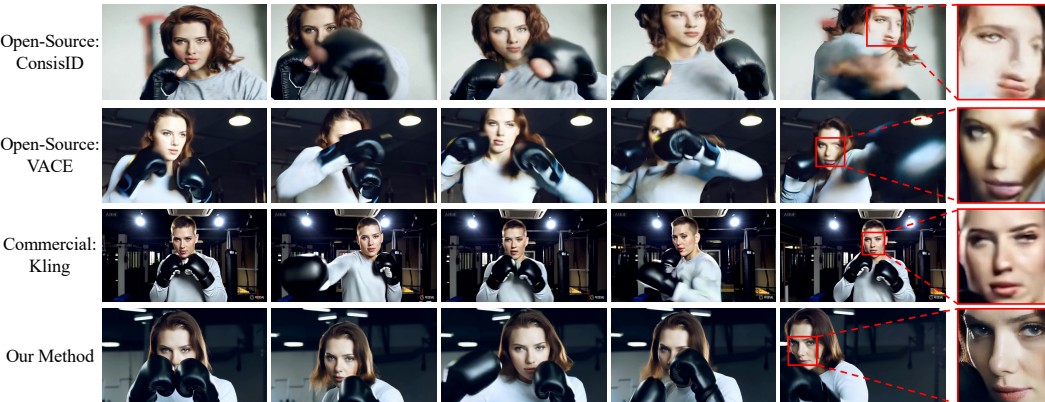

Figure 1: Visualization results from four different IPT2V methods. ConsisID Yuan et al. (2025) shows severe distortion of the facial structure. VACE Jiang et al. (2025) and Kling Kling (2025) suffer from significant distortion of facial identity details. In contrast to these open-source and commercial methods, our method exhibits clear facial structure and high-fidelity identity details as the facial pose changes and occlusions occur. Complete comparisons can be found in Fig. 13.

significant variations in facial pose as well as facial occlusions. We can observe that both open-source and commercial approaches tend to produce noticeable distortion in the facial region as the subject moves and their facial expressions or pose change. This phenomenon may be attributed to the fact that such methods can only capture a single facial pose information from an input reference image, limiting their ability to handle scenarios with significant variations in facial pose. A question arises: *Can we capture global facial pose information from an input single-view reference image?*

In this paper, we propose a pose-faithful facial identity preservation learning framework, named *FaithfulFaces*, to address the aforementioned problem. We first propose a pose-shared identity aligner to encode global facial pose representation from the input single-view reference image. This aligner establishes a pose-shared dictionary to project diverse facial poses into a refined dictionary space, which is learned by a well-crafted pose variation–identity invariance constraint. In this constraint, face images from the same identity but with different poses are treated as positive pairs, while others serve as negative samples. In particular, we incorporate Euler angle embedding learning into the aligner to provide explicit pose cues during the refinement and alignment processes.

Furthermore, to support our FaithfulFaces learning, we design a new dataset collection and processing pipeline that constructs a high-quality, task-specific video dataset with significant facial pose variations to provide a robust training foundation. Finally, the well-trained framework is capable of naturally extracting global facial pose representations as holistic facial priors, enabling foundational generative models to better preserve identity in generated videos. As illustrated in Fig. 1, our method demonstrates superior consistency in maintaining facial identity throughout the generated video as the facial pose changes and occlusions occur. The contributions of this work are threefold:

• We systematically analyze the limitations and potential reasons of existing IPT2V methods in complex facial dynamic scenes, and propose a pose-faithful facial identity preservation learning paradigm, FaithfulFaces, to better preserve consistent identity in generated videos.

• We design a pose-shared identity aligner to encode global facial pose representation from the input single-view reference image via a pose-shared dictionary and a pose variation–identity invariance constraint with Euler angle embedding learning. Additionally, we develop a new dataset pipeline to construct a task-oriented, high-quality video dataset with substantial facial pose diversity to ensure robust model training.

• We perform extensive experiments across diverse identity and dynamic scenarios. Both quantitative and qualitative results demonstrate the effectiveness of our FaithfulFaces, surpassing existing open-source and commercial methods.

## 2 RELATED WORK

Thanks to the powerful data distribution modeling capability and stable training process of the continuous-time generative models Song et al. (2021); Lipman et al. (2023); Liu et al. (2023), large-scale text-to-video generative models Polyak et al. (2024); Kong et al. (2024); Yang et al. (2025); Wan et al. (2025); Gao et al. (2025) have been rapidly developed, further facilitating the Identity-preserving text-to-video generation (IPT2V) task. In the early stage, He *et al.* He et al. (2024) proposed the ID-Animator method that uses a Unet-based lightweight text-to-video model AnimateDiff Guo et al. (2024) and builds a face adapter for IPT2V.

The recent Diffusion Transformer (DiT) architecture Peebles & Xie (2023) has shown promising generative capabilities and has become a mainstream backbone for video generation, such as open-source models HunyuanVideo Kong et al. (2024), CogVideoX Yang et al. (2025), and Wan Wan et al. (2025). Therefore, many recent IPT2V works are built upon and extend the DiT-based models Yuan et al. (2025); Zhang et al. (2025b;a); Xue et al. (2025); Zhong et al. (2025); Fei et al. (2025); Deng et al. (2025). For example, ConsisID Yuan et al. (2025) utilized CogVideoX as the basic generative model and designed a global and local facial extractor to capture global structure and local details as identity information. HunyuanCustom Hu et al. (2025) was built upon the HunyuanVideo foundational model. VACE Jiang et al. (2025), Phantom Liu et al. (2025), SkyReels-A2 Fei et al. (2025), MAGREF Deng et al. (2025), and Stand-In Xue et al. (2025) used Wan as the foundational model.

Furthermore, due to the extremely broad range of real-world applications for IPT2V, numerous successful commercial models and tools have emerged, such as Vidu Vidu (2025), Pika Pika (2025), Kling Kling (2025). However, whether open-source methods or commercial tools, they are difficult to deal with complex facial dynamics, leading to distorted identity information in the generated videos. Therefore, we propose a new FaithfulFaces learning framework to mitigate the above issue.

## 3 METHOD

### 3.1 PROBLEM FORMULATION

**Problem.** Let $I_{\text{ref}}$ and $\mathcal{P}$ denote a reference face image and a text prompt describing the semantics of the target video, respectively. The goal of identity-preserving text-to-video (IPT2V) generation is to create a video $\mathcal{V}$ under the condition of $I_{\text{ref}}$ and $\mathcal{P}$. Thus, $\mathcal{V}$ should satisfy: **i)** the semantic information of $\mathcal{V}$ is aligned with $\mathcal{P}$ (i.e., textual alignment); and **ii)** most importantly, the facial identity information of the subject in $\mathcal{V}$ is consistent with $I_{\text{ref}}$. The generation process can be formalized as:

$$\mathcal{V} = \mathcal{G}\left(\mathbf{Z} \sim \mathcal{N}(\mu, \sigma^2), \phi(I_{\text{ref}}), \mathcal{P}\right), \tag{1}$$

where $\mathcal{G}$ is a text-to-video foundational generative model (e.g., Wan Wan et al. (2025)). $\mathbf{Z}$ is a prior state sampled from the Gaussian prior distribution. $\phi$ denotes a function used to encode the identity information of $I_{\text{ref}}$. For the above equation, the foundational model $\mathcal{G}$ determines the degree of semantic alignment between $\mathcal{V}$ and $\mathcal{P}$. Therefore, researchers only need to select the strongest pretrained model and keep its original prior knowledge (e.g., LoRA Adapter Hu et al. (2022)) during training, which is not the focus of the IPT2V task. For the function $\phi$, which determines the fidelity of facial identity information, i.e., the consistency of facial structure and the fidelity of facial texture details in the generated video $\mathcal{V}$. Thus, this is a critical issue in the IPT2V task, and researchers are dedicated to constructing a robust $\phi$ that accurately represents the subject's identity information.

Recent state-of-the-art works have made various attempts and proposed diverse $\phi$ to improve the performance of IPT2V. For example, ConsisID Yuan et al. (2025) proposed a global facial extractor and a local facial extractor to extract low-frequency structures and high-frequency details of the reference image $I_{\text{ref}}$, respectively. Magic Mirror Zhang et al. (2025a) designed a dual-branch facial feature extractor to capture both identity and structural features. However, they may struggle to handle situations involving complex facial dynamics, such as drastic changes in facial poses and emotions, or facial occlusions, resulting in distorted facial identity and facial structure in the generated videos. The reason behind this phenomenon is that the encoded identity information can only represent a single pose view of the input image, failing to capture faithful global pose information.

**Main Idea.** The identity information encoder could be partitioned into two parts: a basic facial identity encoder $\phi_{\text{bas}}$ and a global facial pose encoder $\phi_{\text{gfp}}$. The former aims to encode the single-

**Training**

**Inference**

Figure 2: The framework of FaithfulFaces. During training, given $n$ input videos per iteration, FaithfulFaces first randomly samples and crops two face images from each video. The cropped face images are then fed into a pose estimator to regress the three Euler angles, e.g., $(\text{pitch}_1^{p_1}, \text{yaw}_1^{p_1}, \text{roll}_1^{p_1})$, where $p_1$ and $p_2$ are simply used to mark two different poses. Next, the predicted Euler angles and the face images are then jointly fed into a pose-shared identity aligner, yielding $2n$ refined facial representations (e.g., $\mathbf{S}_1^{p_1}$), which are utilized to form a pose variation–identity invariance constraint. Finally, the refined representations are injected into the noisy videos as input to the foundational generative model for joint optimization. During inference, FaithfulFaces encodes a global facial pose feature from a single face image and incorporates it into the generative model to produce videos.

view facial structure information and facial texture details as existing methods do, and the latter aims to capture global facial pose representation. Formally, our generation process is defined as:

$$\mathcal{V} = \mathcal{G}\left(\mathbf{Z} \sim \mathcal{N}(\mu, \sigma^2), [\phi_{\text{base}}(I_{\text{ref}}), \phi_{\text{gfp}}(I_{\text{ref}})], \mathcal{P}\right). \tag{2}$$

Accordingly, there are two questions that need to be solved:

- Global facial pose encoder $\phi_{\text{gfp}}$. Representing faithful global facial pose from the input single-view reference image $I_{\text{ref}}$ as introduced in Sec. 3.3.
- Automatic facial video dataset pipeline $P_f$. Collecting and preprocessing the video data with large changes in facial poses for training $\phi_{\text{gfp}}$ as introduced in Sec. 3.4.

## 3.2 OVERVIEW FRAMEWORK

The overview framework of FaithfulFaces is illustrated in Fig .2, which is divided into the training stage and the inference stage. For the training stage, assuming there are $n$ videos as input for each training iteration, we first randomly sample and crop two face images from each video. Subsequently, the cropped face images are fed into a pose estimator to regress the three Euler angles (i.e., pitch, yaw, roll) of the facial pose for each face image. These Euler angles, along with the face images, are then fed into our proposed pose-shared identity aligner to output $2n$ refined facial representations. Furthermore, the $2n$ facial representations from all video samples can be combined into two batches of facial data to form a pose variation–identity invariance constraint. In this constraint, face images from the same identity with different poses are paired as positive samples (diagonal pairs), while those of different identities are paired as negative samples. Finally, the output global facial pose features are injected into the noisy videos as input to the foundational generative model. In practice, we utilize the VACE Jiang et al. (2025) as our foundational model and employ a LoRA training mode to fit these new data, where the VACE blocks are the basic facial identity encoder $\phi_{\text{bas}}$ in Eq. (2) to extract the single-view facial structure information and facial texture details.

During inference, users need only supply a single face image. The pose estimator regresses the Euler angles from this image, and both the angles and the image are passed to a well-trained identity

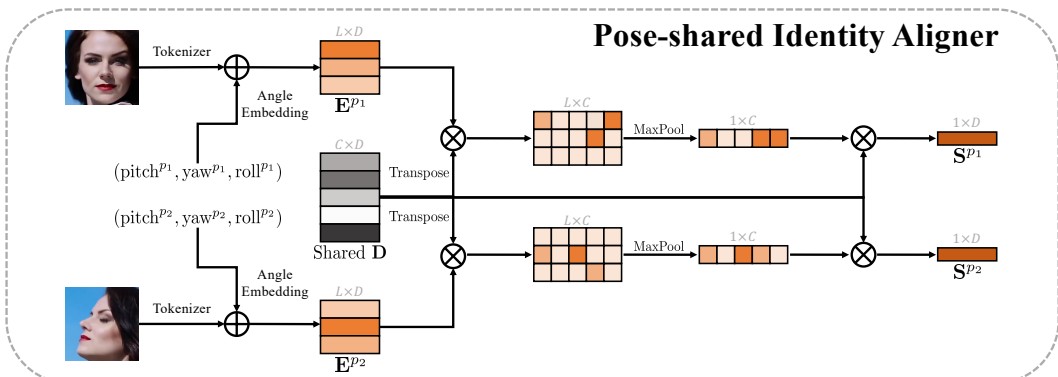

Figure 3: Architecture of the pose-shared identity aligner. Initially, the input face images are tokenized into sequential face embeddings. The corresponding Euler angles are encoded as Euler angle embeddings and injected into the face embeddings, resulting in the combined embeddings $\mathbf{E}^{p_1}$ and $\mathbf{E}^{p_2}$. Then, a pose-shared dictionary $\mathbf{D}$ is employed to refine and align $\mathbf{E}^{p_1}$ and $\mathbf{E}^{p_2}$, yielding the global facial pose representations $\mathbf{S}^{p_1}$ and $\mathbf{S}^{p_2}$. Finally, these representations serve as pose-faithful facial priors for the foundational generative model.

aligner to generate the global facial pose representation. The representation is then incorporated into the noisy video and, in combination with the text prompt and face image, for target video generation.

### 3.3 POSE-SHARED IDENTITY ALIGNER FOR GLOBAL FACIAL POSE REPRESENTATION

For the above framework, the most critical question is how to design and train the pose-shared identity aligner, i.e., encoder $\phi_{\text{gfp}}$ in Eq. (2), to represent robust global facial pose information.

Inspired by the dictionary learning Van Den Oord et al. (2017); Duan et al. (2022), the key of our pose-shared identity aligner is to align the different facial poses into a refined dictionary space. Fig. 3 shows the architecture of the pose-shared identity aligner, which can receive face images with various poses and tokenize them into sequential face embeddings. These vanilla face embeddings contain only implicit pixel-level facial pose information, which hinders the model's ability to perceive facial pose. Thus, we aim to provide explicit pose information to guide the model's representation. Specifically, we utilize a pretrained facial pose estimator (6DRepNet Hempel et al. (2022) in practice) to regress three Euler angles: pitch, yaw, and roll. Notably, Euler angles possess a periodic property, which makes it natural to generate their embeddings using the timestep encoding method employed in diffusion models Ho et al. (2020). As shown in Fig. 3, we inject the Euler angle embeddings into the vanilla face embeddings to generate two new embeddings, marked as $\mathbf{E}^{p_1} \in \mathbb{R}^{L \times D}$ and $\mathbf{E}^{p_2} \in \mathbb{R}^{L \times D}$, where $L$ and $D$ denote the sequence length and dimensionality. $p_1$ and $p_2$ are simply used to mark two different poses. With these embeddings, we then define a learnable pose-shared dictionary matrix $\mathbf{D} \in \mathbb{R}^{C \times D}$, where $C$ indicates the number of dictionary elements. Subsequently, $\mathbf{E}^{p_1} \in \mathbb{R}^{L \times D}$ and $\mathbf{E}^{p_2} \in \mathbb{R}^{L \times D}$ are projected into a dictionary space by calculating the correlation between each face embedding and $\mathbf{D}$ to obtain the correlation matrices, which can be further capsuled into two dictionary weights $\mathbf{W}^{p_1}$ and $\mathbf{W}^{p_2}$ by a max pooling:

$$\mathbf{W}^{p_1} = \text{MaxPool}(\mathbf{E}^{p_1} \otimes \mathbf{D}^\top) \in \mathbb{R}^{1 \times C}, \quad \mathbf{W}^{p_2} = \text{MaxPool}(\mathbf{E}^{p_2} \otimes \mathbf{D}^\top) \in \mathbb{R}^{1 \times C}, \quad (3)$$

where $\text{MaxPool}(\cdot)$ denotes a max pooling operation and $\otimes$ means matrix multiplication. Finally, these dictionary weights can be used to obtain the global facial pose representations $\mathbf{S}^{p_1}$ and $\mathbf{S}^{p_2}$:

$$\mathbf{S}^{p_1} = (\mathbf{W}^{p_1} \otimes \mathbf{D}) \in \mathbb{R}^{1 \times D}, \quad \mathbf{S}^{p_2} = (\mathbf{W}^{p_2} \otimes \mathbf{D}) \in \mathbb{R}^{1 \times D}. \quad (4)$$

To optimize this aligner, we observe that the two batches of input facial data with different poses can exactly form a CLIP-like contrastive paradigm, as shown in the upper part of Fig. 2. Therefore, we apply the most commonly used contrastive learning loss Radford et al. (2021) to train our aligner:

$$\mathcal{L}_{\text{PIA}} = -\frac{1}{n}\sum_{i=1}^{n} \log \frac{\exp(\text{sim}(\mathbf{S}_i^{p_1}, \mathbf{S}_i^{p_2})/\tau)}{\sum_{j=1}^{n} \exp(\text{sim}(\mathbf{S}_i^{p_1}, \mathbf{S}_j^{p_2})/\tau)} - \frac{1}{n}\sum_{i=1}^{n} \log \frac{\exp(\text{sim}(\mathbf{S}_i^{p_2}, \mathbf{S}_i^{p_1})/\tau)}{\sum_{j=1}^{n} \exp(\text{sim}(\mathbf{S}_i^{p_2}, \mathbf{S}_j^{p_1})/\tau)}, \quad (5)$$

where $n$ is the number of matched identity pairs in each training mini-batch, $\text{sim}(\cdot, \cdot)$ denotes the cosine similarity function, and $\tau$ is a learnable temperature parameter with the default setting of

Figure 4: Dataset collection and processing pipeline. In Step 1, videos without face or with multiple faces are filtered out. Step 2 aims to select videos that exhibit significant variations in facial pose. For Step 3, we generate a text prompt for each selected video using MLLM. Step 4 ultimately integrates these fragmented data into a cohesive whole.

Radford et al. (2021). During the whole training process, we integrate $\mathcal{L}_{\text{PIA}}$ with the objective of the generative model (i.e., flow matching Liu et al. (2023)) $\mathcal{L}_{\text{FM}}$ to reach the full optimization objective:

$$\mathcal{L}_{\text{total}} = \mathcal{L}_{\text{PIA}} + \mathcal{L}_{\text{FM}}. \tag{6}$$

In practice, $\mathcal{L}_{\text{PIA}}$ and $\mathcal{L}_{\text{FM}}$ are responsible for their respective tasks during the training process. $\mathcal{L}_{\text{PIA}}$ is dedicated to constraining the alignment of different poses, while $\mathcal{L}_{\text{FM}}$ is dedicated to constraining the LoRA parameters to adapt to the input's global facial pose representation. This approach ensures that the different loss functions can focus on handling their specific tasks.

**Remark 1** (**Deep insights and observations**) *Our design of the pose-shared identity aligner is not only intuitive but also admits a theoretical justification. Recall that $\mathcal{L}_{\text{PIA}}$ is equivalent to the InfoNCE loss Oord et al. (2018), which provides a lower bound of the mutual information:*

$$I(\mathbf{S}^{p_1}; \mathbf{S}^{p_2}) \geq \log(n) - \mathcal{L}_{\text{PIA}}. \tag{7}$$

*This inequality implies that minimizing $\mathcal{L}_{\text{PIA}}$ is not only aligning pose-variant embeddings but also maximizing the shared identity information across different poses. Hence, our aligner has an information-theoretic guarantee: the learned global representation cannot collapse unless $I(\mathbf{S}^{p_1}; \mathbf{S}^{p_2})$ vanishes. From the experimental observation, the visualization of the encoded facial identity in Fig. 6 confirms the above insights. Furthermore, the learned dictionary reveals meaningful activation patterns, wherein images with similar poses tend to frequently activate particular dictionary elements, as illustrated in Fig. 7. This indicates that the learned dictionary facilitates robust representation of faces across a wide range of poses.*

### 3.4 DATASET CONSTRUCTION

Beyond framework design, a critical challenge persists: constructing a video dataset with significant variations in facial poses for training our proposed pose-shared identity aligner. This is because ordinary facial micro-movements or static videos are insufficient to satisfy our training requirements.

To address this issue, we construct a new dataset collection and processing pipeline. Note that this part omits standard data collection and preprocessing procedures that have been widely adopted in previous works Jiang et al. (2025); Hu et al. (2025); Liu et al. (2025), such as video clip segmentation, resolution standardization, OCR filter, aesthetic filter, clarity filter, etc. The original videos are from the internet and in-house sources, and the resolution of each video is standardized to $832 \times 480$ pixels. Fig. 4 illustrates our dataset collection and processing pipeline, which consists of four steps: face detection, pose estimation, video prompt generation, and processed data combination.

**Face Detection.** Since our work only focuses on the single-subject video generation task, we first need to filter out two types of videos: videos without face and videos with multiple faces. Specifically, we utilize InsightFace InsightFace (2025) for face detection on each video frame. Videos are

filtered out if more than two faces are detected in any single frame. Additionally, videos in which no faces are detected throughout the entire sequence are also excluded.

**Pose Estimation.** This step constitutes the core of the entire dataset pipeline, aiming to select videos that exhibit significant variations in facial pose. Taking a video $\mathcal{V}$ as an example, we first use the facial bounding boxes obtained in the previous step to crop the face regions from each video frame. These cropped face regions are then fed into the pose estimator 6DRepNet to predict three Euler angles for each detected face. Note that in practice, we enlarge the bounding boxes by a factor of 1.5 to predict Euler angles more accurately. Next, the three Euler angles for each face are stored separately in three lists, denoted as $\mathcal{X}_{\text{pitch}}$, $\mathcal{X}_{\text{yaw}}$, and $\mathcal{X}_{\text{roll}}$, and we can calculate the variation of Euler angles throughout the entire video:

$$\text{Var} = [\max(\mathcal{X}_{\text{pitch}}) - \min(\mathcal{X}_{\text{pitch}})] + [\max(\mathcal{X}_{\text{yaw}}) - \min(\mathcal{X}_{\text{yaw}})] + [\max(\mathcal{X}_{\text{roll}}) - \min(\mathcal{X}_{\text{roll}})], \quad (8)$$

where $\max(\cdot)$ and $\min(\cdot)$ represent the maximum and minimum values in the list, respectively. Furthermore, it is necessary to determine a reliable variation threshold to filter out qualified videos. To determine this threshold, we first randomly sample 2000 videos from the output of step 1 and manually annotate them. Our criterion for qualified videos is that the facial pose in the video must show at least a transition from frontal to profile (or vice versa), or exhibit significant up-and-down movement. Videos meeting these criteria are labeled as qualified, and we finally determine that the threshold is 120. With this threshold, we can filter out videos with large facial pose changes; that is, $\text{Var} > 120$ is qualified, while $\text{Var} < 120$ is discarded.

**Video Prompt Generation.** After collecting qualified videos, we need to generate a text prompt for each video. Here, we use Qwen2.5-VL Bai et al. (2025) to generate information-rich text prompts for qualified videos, focusing on describing the subjects' appearance, actions, and background. We then perform extensive manual calibration and refinement to improve the accuracy of text prompts.

**Processed Data Combination.** After the aforementioned three steps of data screening and preprocessing, we ultimately integrate these fragmented data into a cohesive whole. As shown in step 4 of Fig. 4, each sample in our well-constructed dataset contains four elements: video, text prompt, cropped face images, and Euler angles. We manually check all processed data to ensure that all videos are qualified enough, and ultimately generate 51,624 samples for model training.

## 4  EXPERIMENTS

### 4.1  IMPLEMENTATION DETAILS

Our FaithfulFaces framework utilizes the DiT-based generative model VACE-14B Jiang et al. (2025) as our foundational model. For the pose-shared identity aligner, the number of dictionary elements of $\mathbf{D}$ is set to 4096, which is determined by experiments (Appendix A.2). We set the resolution of each video to $832 \times 480$ pixels and extract 81 consecutive frames for training. In the training phase, we use the LoRA training mode with rank 128 (applied only to the QKV projections in each block) to fit new data. The whole framework is trained on 32 NVIDIA H20 GPUs with a batch size of 32. In addition, we set an independent batch size of 1024 for the pose-shared identity aligner to perform adequate pose alignment, and the total number of training steps is set to 5000.

**Evaluation details.** We conduct experiments and evaluations on several face images used in ConsisID Yuan et al. (2025), which consist of 30 persons, and we randomly sample one image for each identity. We then construct 20 challenging text prompts that drive the models to generate videos featuring significant facial pose variations, expression changes, and facial occlusions across diverse scenarios. The details can be found in Appendix A.8. We consider four standard evaluation metrics that are used in prior works Polyak et al. (2024); Yuan et al. (2025) to measure the quality of generated videos. **FaceSim-Arc** and **FaceSim-Cur** are employed to assess identity preservation by measuring feature discrepancies between face regions in the generated videos and those in real face images within the ArcFace Deng et al. (2019) and CurricularFace Huang et al. (2020) feature spaces. For visual quality, we utilize the commonly used **FID** Heusel et al. (2017) by calculating feature differences in the face regions between the generated frames and real face images within the InceptionV3 Szegedy et al. (2016) feature space. For textual alignment, we use the **CLIPScore** Hessel et al. (2021) to measure the similarity between the generated videos and the text prompts.

Input Reference Image and Text Prompt

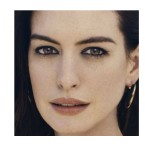

*The video features a person exercising on a sunlit, open grassy field, wearing a light gray long-sleeve shirt and black boxing gloves.* The individual is engaged in a boxing routine, demonstrating various punches and defensive maneuvers...

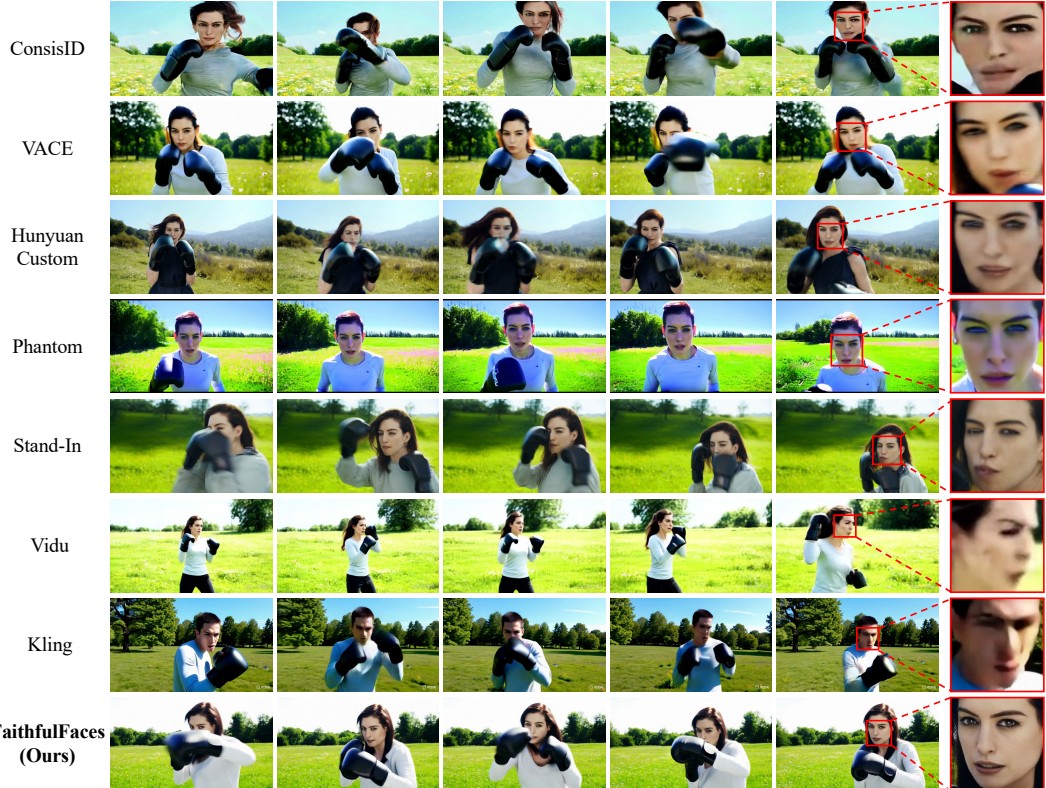

Figure 5: Visual comparisons of different methods. The goal is to generate a video of a person engaging in a boxing routine, characterized by diverse facial poses and instances of facial occlusion. In contrast to these state-of-the-art methods, FaithfulFaces produces a video of superior quality, exhibiting clear facial structures and consistent identity preservation.

## 4.2 BASELINE COMPARISONS

We compare our FaithfulFaces with the current state-of-the-art methods, including two commercial products (Vidu Vidu (2025), Kling Kling (2025)) and eight open-source models (ConsisID Yuan et al. (2025), VACE Jiang et al. (2025), HunyuanCustom Hu et al. (2025), Stand-In Xue et al. (2025), Phantom Liu et al. (2025), Concat-ID-Wan Zhong et al. (2025), SkyReels-A2 Fei et al. (2025), MAGREF Deng et al. (2025) ). For these open-source methods, ConsisID is based on CogVideoX-5B Yang et al. (2025), HunyuanCustom is based on HunyuanVideo Kong et al. (2024), VACE, Stand-In, Phantom, Concat-ID-Wan, SkyReels-A2, and MAGREF are based on Wan Wan et al. (2025), providing diversity for evaluation and comparison. For each method, we generate 600 videos (30 persons × 20 prompts) for evaluation. Below, we analyze quantitative and qualitative experiments.

**Quantitative results.** Tab. 1 lists the quantitative results of different methods. From these results, we can observe that FaithfulFaces achieves the best IPT2V performance under four evaluation metrics. In particular, FaithfulFaces gains considerable performance improvements in the FaceSim-Cur and FaceSim-Arc metrics used to measure identity preservation of generated videos. This improvement can be attributed to FaithfulFaces's ability to provide a robust global facial pose prior for foundational generative models, enabling more effective identity preservation in generated videos.

**Qualitative results.** Fig. 5 provides some visual comparisons of our FaithfulFaces against seven baselines, including a case of generating a video of a person engaging in a boxing routine. We can first observe that the subjects in the videos generated by different methods have various facial pose

Table 1: Quantitative comparison with evaluated baselines.

| Methods | Open-Source | FaceSim-Cur ↑ | FaceSim-Arc ↑ | FID ↓ | CLIPScore ↑ |
|---|---|---|---|---|---|
| Vidu Vidu (2025) | ✗ | 0.293 | 0.278 | 234.65 | 30.08 |
| Kling Kling (2025) | ✗ | 0.447 | 0.416 | 194.80 | 33.06 |
| ConsisID Yuan et al. (2025) | ✓ | 0.365 | 0.350 | 205.03 | 30.29 |
| VACE-14B Jiang et al. (2025) | ✓ | 0.403 | 0.382 | 191.02 | 31.83 |
| HunyuanCustom Hu et al. (2025) | ✓ | 0.453 | 0.432 | 187.32 | 31.36 |
| Stand-In Xue et al. (2025) | ✓ | 0.415 | 0.395 | 196.21 | 30.38 |
| Phantom-14B Liu et al. (2025) | ✓ | 0.484 | 0.456 | 214.99 | 29.67 |
| Concat-ID-Wan Zhong et al. (2025) | ✓ | 0.408 | 0.387 | 189.55 | 31.49 |
| SkyReels-A2 Fei et al. (2025) | ✓ | 0.410 | 0.384 | 237.29 | 28.10 |
| MAGREF Deng et al. (2025) | ✓ | 0.417 | 0.392 | 207.69 | 31.13 |
| **FaithfulFaces (Ours)** | ✓ | **0.568** | **0.542** | **164.24** | **33.93** |

changes and even facial occlusion due to intense movements. Furthermore, we discover that these open-source and commercial methods exhibit varying degrees of identity distortion and facial collapse as the subject moves. In contrast, our FaithfulFaces can output a high-quality video with clear facial structures and consistent identity details. Similar observations are made in Appendix A.12, where more visual results are provided. Video demos are included in the Supplementary Material.

### 4.3 ABLATION STUDIES

We evaluate the effects of the key components in FaithfulFaces, including the pose-shared identity aligner (Identity Aligner) and the injection of Euler angle embeddings (Euler Embedding). The results are presented in Tab. 2, from which we draw the following conclusions: **i)** Identity Aligner is effective and yields substantial performance improvements, as it represents global facial pose information from the input single-view reference image, thereby enhancing the identity consistency of the generated videos. **ii)** The inclusion of Euler Embedding yields further improvements, confirming the feasibility and effectiveness of explicitly injecting pose information. More ablation studies are provided in Appendix A.2 and A.3, including qualitative analysis and dictionary elements.

Table 2: Ablation study of the key components in FaithfulFaces.

| Identity Aligner | Euler Embedding | FaceSim-Cur ↑ | FaceSim-Arc ↑ | FID ↓ | CLIPScore ↑ |
|---|---|---|---|---|---|
| ✓ | ✓ | **0.568** | **0.542** | **164.24** | **33.93** |
| ✓ | ✗ | 0.522 | 0.497 | 173.57 | 33.31 |
| ✗ | ✗ | 0.437 | 0.414 | 186.82 | 32.05 |

**Visualization of the encoded facial identity.** Fig. 6 visualizes the distribution of encoded facial identities under different settings. Specifically, we randomly select 7 videos with different identities that are not included in the training data, and sample 8 face images with different facial poses from each video. All sampled face images are then encoded by the different methods, and the encoded features are projected into a 2D space by t-SNE Maaten & Hinton (2008). We can observe that *FaithfulFaces w/o (Identity Aligner, Euler Embedding)* suffer from the collapse of facial identity due to the absence of global facial pose awareness. At the same time, *FaithfulFaces w/o (Euler Embedding)* alleviates the collapse of facial identity, but the discriminability of its facial identity representation remains limited. In contrast, *FaithfulFaces* shows promising identity separability, demonstrating its faithful facial identity and naturally enhancing the performance of IPT2V.

## 5 CONCLUSION

In this paper, we have proposed FaithfulFaces, a pose-faithful facial identity preservation learning framework for IPT2V. FaithfulFaces is motivated by the observation that existing methods often struggle to handle some intricate facial dynamic scenarios, largely due to their insufficient awareness of global facial pose. To encode the global facial pose representation from the input single-view face image, we propose a pose-shared identity aligner that refines and aligns distinct facial poses by a pose-shared dictionary and a pose variation–identity invariance constraint with Euler angle embedding learning. In particular, we construct a task-oriented, high-quality dataset with substantial facial pose diversity for robust training. Extensive experiments validate the effectiveness of FaithfulFaces.

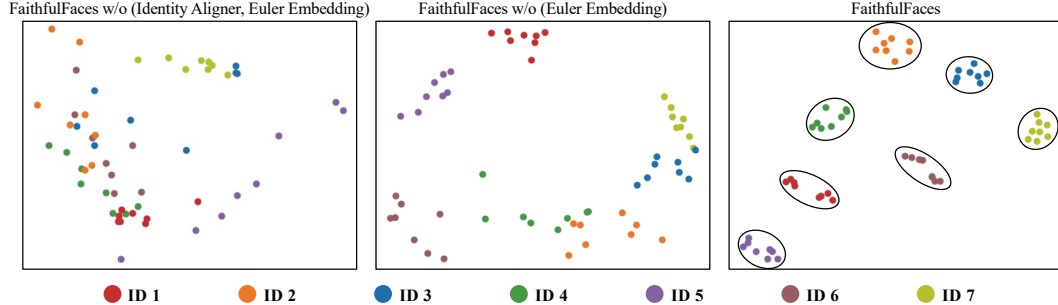

Figure 6: Visualization of the encoded facial identity (ID). FaithfulFaces demonstrates promising ID separability, indicating that its encoded identity representation exhibits high faithfulness and fidelity.

**Limitations and discussion.** In the experiments, we observe some failure cases. For example, when the user provides an input image with severe facial occlusion, it leads to a reduction in the identity consistency of the generated video. The reason for this is that images with severe facial occlusion result in a serious loss of facial structure and facial texture information. Please note that this scenario is an open issue for the IPT2V field, and we believe this problem will be gradually alleviated in future research.

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

# A Appendix

## A.1 Observations in Learned Dictionary

To explicitly observe the learned pose-shared dictionary, we visualize the activations of the dictionary for five representative facial poses in Fig. 7. Specifically, we screen all face images of five facial poses in our dataset based on Euler angles and feed them into the pose-shared identity aligner to calculate the dictionary weight corresponding to each face image. We then record the indices of the top-10 elements in each weight vector, representing the most prominently activated dictionary elements for each face image. From the results presented in Fig. 7, we observe that particular dictionary elements are consistently activated by face images with similar poses. For example, the *frontal pose* tends to activate dictionary elements with indices 3, 562, and 2806, whereas the *upward-looking pose* frequently activates those with indices 2, 704, and 1856. This observation demonstrates that the learned dictionary captures meaningful patterns, potentially enabling robust representations of faces exhibiting a wide range of poses.

| Indices of Dictionary Element | Facial Poses | Examples of Face Images |
|---|---|---|
| 3, 562, 2806 | Frontal Pose
$-25° < $ pitch $ < +25°$
$-25° < $ yaw $ < +25°$
$-25° < $ roll $ < +25°$ | |
| 4, 2264 | Profile (left)
$-25° < $ pitch $ < +25°$
$+30° < $ yaw $ < +90°$
$-25° < $ roll $ < +25°$ | |
| 851, 1954 | Profile (right)
$-25° < $ pitch $ < +25°$
$-30° < $ yaw $ < -90°$
$-25° < $ roll $ < +25°$ | |
| 2, 704, 1856 | Upward-looking
$+30° < $ pitch $ < +90°$
$-25° < $ yaw $ < +25°$
$-25° < $ roll $ < +25°$ | |
| 1358, 3267 | Downward-looking
$-30° < $ pitch $ < -90°$
$-25° < $ yaw $ < +25°$
$-25° < $ roll $ < +25°$ | |

Figure 7: Visualization of learned pose-shared dictionary. We visualize five representative facial poses and observe that similar poses tend to frequently activate particular dictionary elements.

## A.2 EXPLORING THE EFFECTS OF DIFFERENT DICTIONARY ELEMENTS

In this section, we conduct the ablation studies to explore the effects of different dictionary elements in $\mathbf{D}$. Specifically, we conduct six sets of experiments in which the number of dictionary elements is set to 1024, 2048, 4096, 8192, 16384, and 32768, respectively. Fig. 8 illustrates the performance of our FaithfulFaces with various dictionary elements under two FaceSim metrics, and we can observe that the best performance is reached when the number of dictionary elements is set to 4096. Subsequently, the performance exhibits only slight variations as the number of dictionary elements increases further. In addition, the quantitative results of different dictionary elements are listed in Tab. 3. Hence, we use 4096 dictionary elements in our experiments.

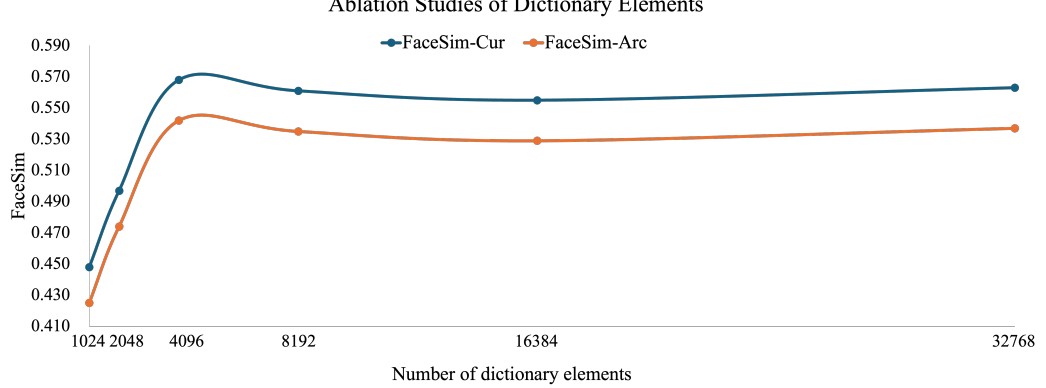

Figure 8: Ablation studies on different dictionary elements.

## A.3 QUALITATIVE ANALYSIS OF ABLATION STUDIES FOR KEY COMPONENTS

In addition to the quantitative results of the ablation studies presented in Tab. 2, we provide a qualitative analysis with visualizations in Fig. 9. We can observe that *Ours w/o (Identity Aligner, Euler*

Table 3: Quantitative results of different dictionary elements.

| Dictionary Elements | FaceSim-Cur ↑ | FaceSim-Arc ↑ | FID ↓ | CLIPScore ↑ |
|---|---|---|---|---|
| 1024 | 0.448 | 0.425 | 183.39 | 32.28 |
| 2048 | 0.497 | 0.474 | 176.80 | 32.89 |
| 4096 | **0.568** | **0.542** | **164.24** | **33.93** |
| 8192 | 0.561 | 0.535 | 165.94 | 33.82 |
| 16384 | 0.555 | 0.529 | 166.64 | 33.83 |
| 32768 | 0.563 | 0.537 | 165.96 | 33.82 |

*Embedding)* shows obvious distortion of facial structures and facial details due to the lack of global facial pose awareness. *Ours w/o Euler Embedding* mitigates facial distortions owing to the global facial pose representation provided by our pose-shared identity aligner; however, its identity consistency and facial stability remain suboptimal. In contrast, *Ours* generates high-quality results with clear facial structures and consistent identity details.

Input Reference Image and Text Prompt 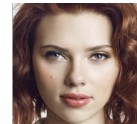 The video features a person in a gym setting, wearing a light gray long-sleeve shirt and black boxing gloves. The individual is engaged in a boxing routine, demonstrating various punches and defensive maneuvers...

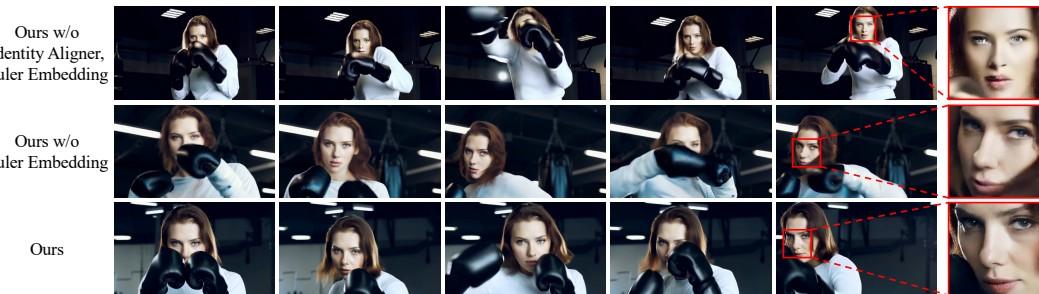

Ours w/o Identity Aligner, Euler Embedding

Ours w/o Euler Embedding

Ours

Figure 9: Visual comparisons of ablation studies for key components in FaithfulFaces.

## A.4 ABLATION STUDIES OF POOLING OPERATION TYPE

We evaluate the effects of different pooling operation types in the pose-shared identity aligner, and the results are listed in Tab. 4. From these results, we can observe that performance is optimal when using the max pooling operation. We also analyzed the potential underlying reason: the majority of information in face images of the same identity across different poses is similar or redundant. Therefore, using max pooling can alleviate a large amount of redundant information and extract highly abstract pose variations.

Table 4: Quantitative results of different pooling operation types.

| Pooling Type | FaceSim-Cur ↑ | FaceSim-Arc ↑ | FID ↓ | CLIPScore ↑ |
|---|---|---|---|---|
| Sum Pooling | 0.444 | 0.421 | 185.40 | 32.17 |
| Mean Pooling | 0.559 | 0.533 | 165.64 | 33.84 |
| Max Pooling | **0.568** | **0.542** | **164.24** | **33.93** |

## A.5 STABILITY OF CONTRASTIVE LOSS IN OPTIMIZATION PROCEDURE

Fig. 10 illustrates the value of the contrastive loss in the pose-shared identity aligner from different training steps. We can observe that the loss value gradually decreases during training and eventually converges to approximately 0.2. These results demonstrate the stability and convergence of contrastive learning for the pose-shared identity aligner during the training process.

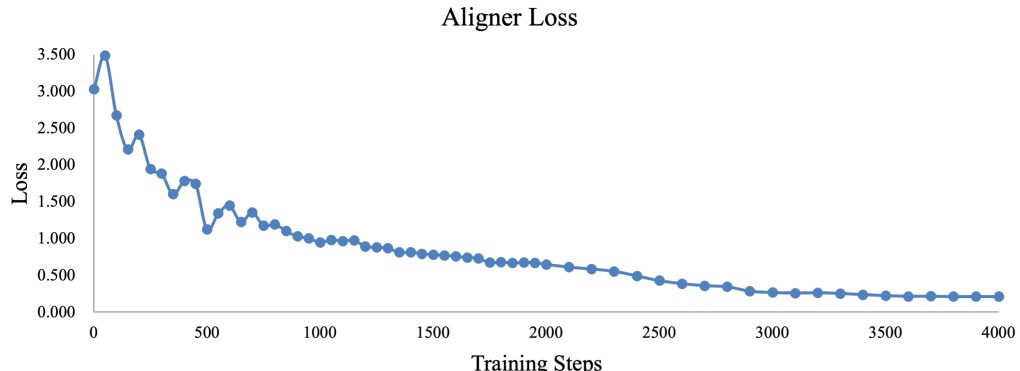

Figure 10: The value of the contrastive loss in the pose-shared identity aligner (named aligner loss) during the training process.

### A.6 DISCUSSION ON NON-FRONTAL VIEW ROBUSTNESS

In this section, we discuss the robustness of the method when the input reference image is a non-frontal view. Specifically, we collect 10 identities with both frontal and non-frontal face images for ablation and comparative (strongest baseline Phantom Liu et al. (2025) as the representative) experiments. The results are shown in the Tab. 5, we can observe that both Ours w/o Aligner and the strongest baseline Phantom suffer a severe performance decrease exceeding 50% when non-frontal face images are used as input. In contrast, our method is able to control the performance decrease within 25%. These results provide strong evidence that the pose-shared identity aligner can mitigate performance degradation in non-frontal face scenarios.

Additionally, we further provide visualization results of the frontal view and the non-frontal view in Fig. 11. We can further observe that when using a non-frontal image as input, the faces generated by the Phantom and Ours w/o Aligner completely collapse. In contrast, our method still maintains identity consistency. These visual results once again demonstrate that our method is capable of improving identity consistency in non-frontal face scenarios.

Table 5: Quantitative results of frontal view and non-frontal view. The values reported in each cell denote FaceSim-Cur/ FaceSim-Arc

| Methods | Frontal | Non-frontal |
|---|---|---|
| Phantom Liu et al. (2025) | 0.470 / 0.435 | 0.231 (↓ 50.85%) / 0.216 (↓ 50.34%) |
| Ours w/o Aligner | 0.448 / 0.423 | 0.202 (↓ 54.91%) / 0.197 (↓ 53.43%) |
| Ours | 0.544 / 0.522 | 0.409 (↓ 24.82%) / 0.392 (↓ 24.90%) |

### A.7 DISCUSSION ON THE ROBUSTNESS OF IDENTITY ALIGNER FOR EULER ANGLES

In our pose-shared identity aligner, the sparsity design of the dictionary representation mechanism inherently possesses a certain degree of noise tolerance. To demonstrate this, we conduct the ablation experiments involving Euler angle perturbations. Specifically, given four perturbation ranges for Euler angles: $-5° \sim +5°$, $-10° \sim +10°$, $-15° \sim +15°$, $-20° \sim +20°$. We apply random perturbations within these ranges to the predicted Euler angles to observe the impact on performance. The experimental results are shown in the Tab. 6, we can observe that performance exhibits only minor variations within the perturbation range of $-15° \sim +15°$. Significant performance degradation occurs only when perturbations exceed $15°$ (representing substantial errors). These results prove that our method exhibits high robustness to Euler angle errors within a certain range.

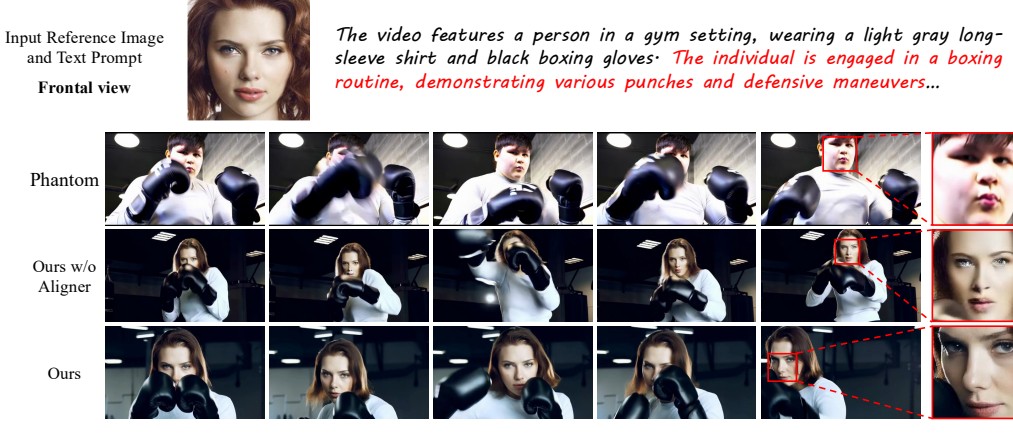

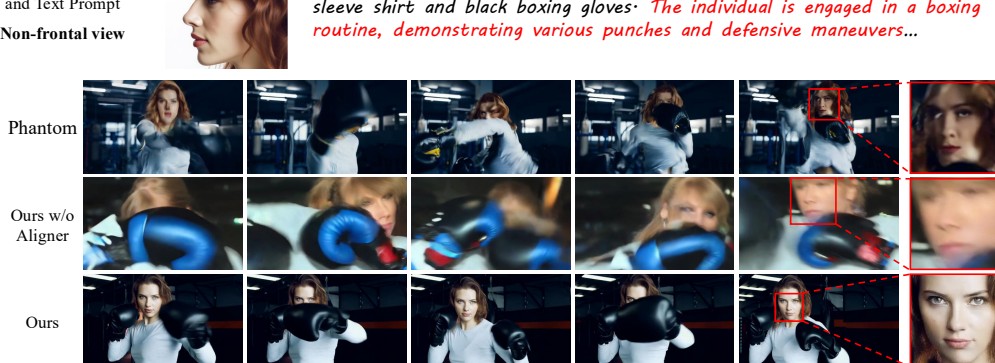

Figure 11: Visual comparisons of frontal view and non-frontal view. We can observe that when using a non-frontal image as input, the faces generated by the SOTA method Phantom and the baseline method (Ours w/o Aligner) completely collapse. In contrast, our method still maintains identity consistency.

Table 6: Quantitative results under different Euler angle perturbations.

| Perturbation Range | FaceSim-Cur ↑ | FaceSim-Arc ↑ | FID ↓ | CLIPScore ↑ |
|---|---|---|---|---|
| No Perturbation | 0.568 | 0.542 | 164.24 | 33.93 |
| $-5° \sim +5°$ | 0.566 | 0.540 | 164.54 | 33.89 |
| $-10° \sim +10°$ | 0.557 | 0.531 | 166.06 | 33.77 |
| $-15° \sim +15°$ | 0.552 | 0.526 | 167.07 | 33.74 |
| $-20° \sim +20°$ | 0.523 | 0.499 | 172.54 | 33.56 |

## A.8 PROMPT CONSTRUCTION

We now elaborate on the construction of challenging test text prompts designed to drive models to generate videos exhibiting significant facial pose variations, expression changes, and facial occlusions across diverse scenarios. For character movement, we select several representative scenes: 1) **Boxing** with facial pose variations and facial occlusions; 2) **Head shaking and dancing** with facial pose variations; 3) The character **transitions from having their back to the camera to facing the camera**; 4) **Ballet** with facial pose variations; 5) **Speech** with facial pose variations and expression changes; 6) Some descriptions used to generate **dramatic changes in facial expressions and poses**, as shown in the third case in Fig. 17.

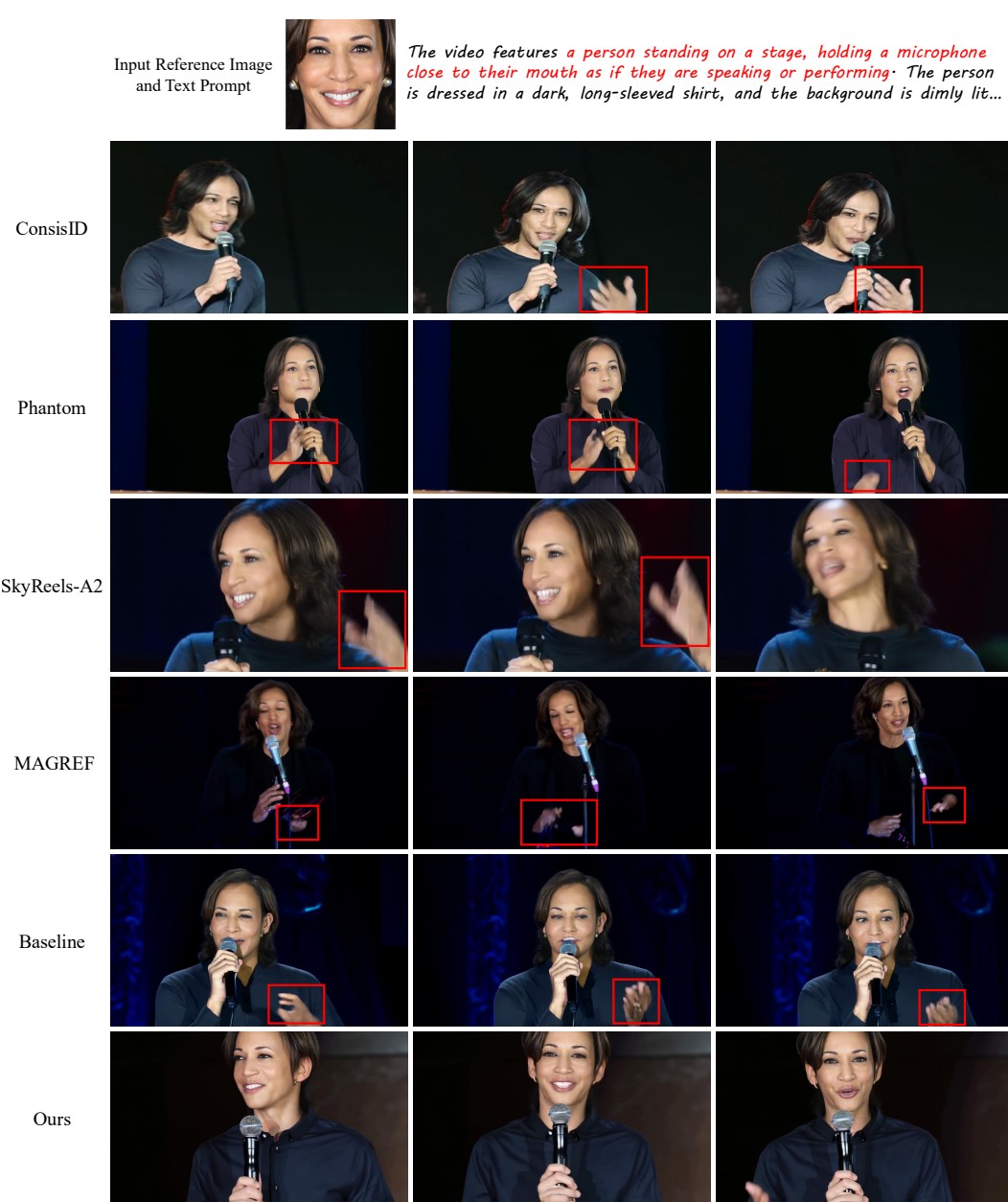

Figure 12: Visual comparisons of different methods. Severe hand distortion and collapse are marked with red boxes. We can observe that unnaturalness, distortion, and even collapse in the hand region are widespread in existing methods.

With these basic movement scenes, we use GPT-4.1[1] to generate information-rich text prompts. Taking the boxing scene as an example, the generated text prompt is: *"The video features a person in a gym setting, wearing a light gray long-sleeve shirt and black boxing gloves. The individual is engaged in a boxing routine, demonstrating various punches and defensive maneuvers. The camera closely follows the person's movements, keeping their face and gloves prominent in the frame, and capturing detailed facial expressions and dynamic action. The background is dimly lit, with overhead lights providing illumination. The gym environment is evident from the visible equipment and the industrial setting, which adds to the intensity of the scene."*. Subsequently, based on this, we employ GPT-4.1 to generate text prompts for different background scenarios, such as *"open grassy*

[1]https://openai.com/index/gpt-4-1/

*field"* in Fig. 13, *"urban street setting"* in Fig. 19, and *"cozy living room"* in Fig. 20. Ultimately, we curate 20 high-quality text prompts that span various actions and background scenarios.

### A.9 ETHICS STATEMENT AND BROADER IMPACT

Advancements in identity-preserving text-to-video generation technology are poised to support and empower the creative processes of artists and designers. FaithfulFace is capable of generating high-quality, realistic human videos. However, it also raises concerns regarding misinformation, potentially undermining the reliability of video content. Additionally, this technology could be misused to generate deceptive content for fraudulent purposes. It is important to recognize that any technology is susceptible to misuse. Nevertheless, it is feasible to train a classifier that can distinguish between real and FaithfulFaces-generated videos based on their texture features.

### A.10 REPRODUCIBILITY STATEMENT

First, we have explained the implementation of FaithfulFaces in detail in Sec. 4.1. Second, we have explained the details of training and inference in Fig. 2 and Sec. 3.2. Third, we have explained the details of the dataset construction in Sec. 3.4. Finally, the code and dataset pipeline used in this work will be open-source online.

### A.11 THE USE OF LARGE LANGUAGE MODELS

This submission utilizes a large language model for grammar checking.

### A.12 MORE VISUALIZATION RESULTS

In this section, we provide more visual comparisons of different methods in Figs. 13, 14, 15, and 16 to demonstrate the effectiveness of our method. Additionally, we provide more showcases of identity-preserving videos generated by our FaithfulFaces in Figs. 17, 18, 19, and 20, covering a variety of identities, actions, and scenes.

Figure 13: Complete visual comparisons of different methods for the case of Fig. 1.

Input Reference Image and Text Prompt

*The video features a person wearing headphones and a black outfit, standing in a park surrounded by tall trees...* *The person is seen with their arms raised above their head, appearing to be dancing or moving rhythmically to music...*

ConsisID

VACE

Hunyuan Custom

Phantom

Stand-In

Vidu

Kling

**FaithfulFaces (Ours)**

Figure 14: More visual comparisons of different methods.

Figure 15: More visual comparisons of different methods.

Figure 16: More visual comparisons of different methods.

Input Reference Image and Text Prompt

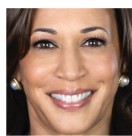

*The video features* *a ballerina performing a graceful dance* *move in an outdoor garden bathed in soft early morning sunlight... The ballerina is wearing a black leotard with long sleeves that extend to her wrists, and her hair is neatly tied back in a bun...*

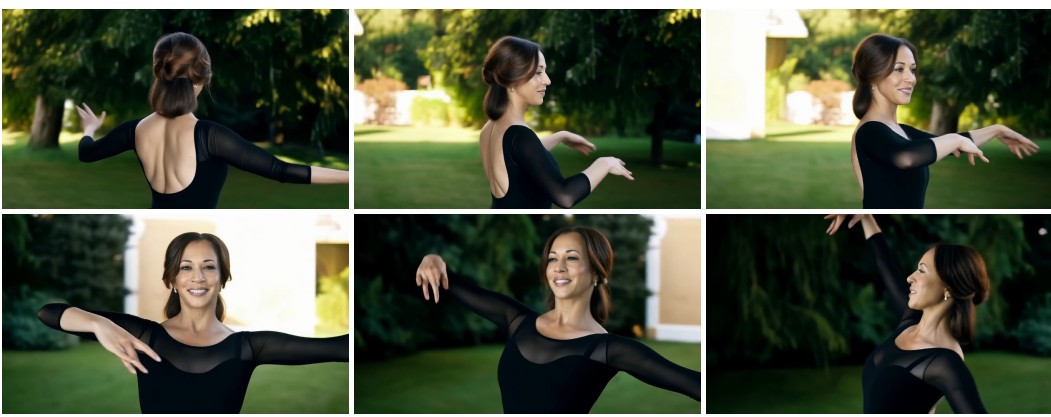

*The video features* *a person standing on a stage, holding a microphone close to their mouth as if they are speaking or performing. The person is dressed in a dark, long-sleeved shirt, and the background is dimly lit...*

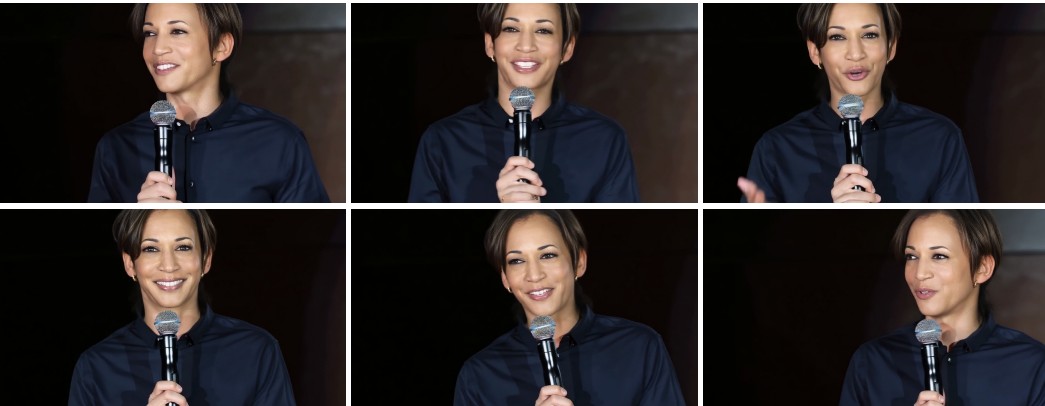

*On a park bench, sunlight filters through the leaves onto a person's face. The person is dressed in casual sportswear and holding a book. At the beginning of the video,* *they are looking down at the book with a focused expression. Suddenly, they look up with surprise, then shift to a puzzled look, followed by laughter...*

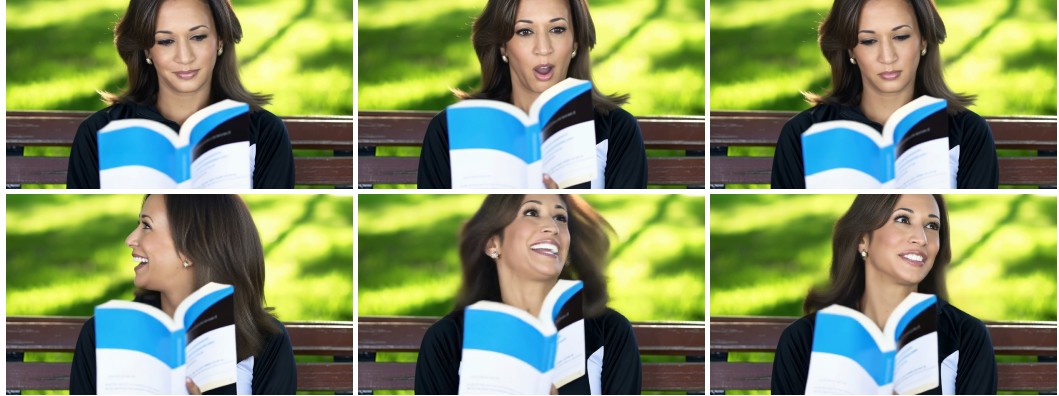

Figure 17: More showcases of identity-preserving videos generated by our FaithfulFaces.

Input Reference Image and Text Prompt

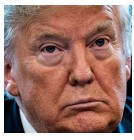

*The video is shot on a city rooftop at night... The person, wearing a black jacket and dark jeans, stands at the edge of the rooftop with a blurred cityscape in the background· At the beginning, the person has their back to the camera and slowly turns to face it...*

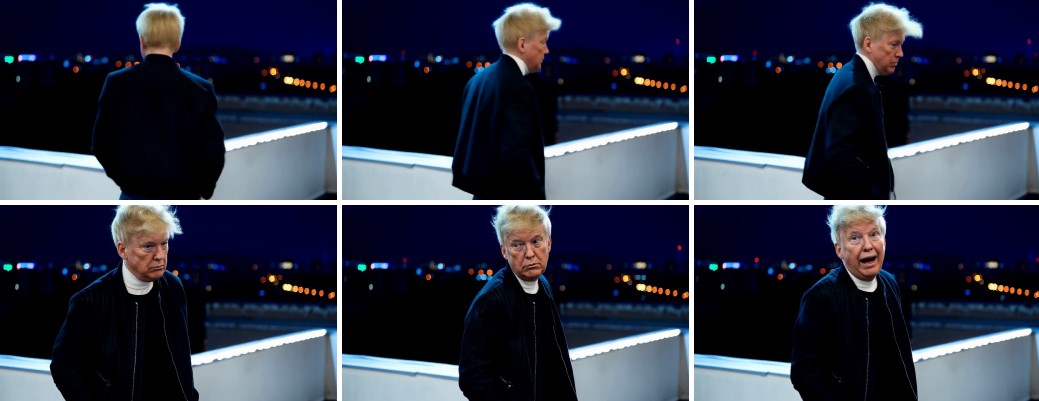

*The video features a person wearing headphones and a black outfit, standing near a waterfront promenade with boats docked nearby... The person is seen with their arms raised above their head, appearing to be dancing or moving rhythmically to music...*

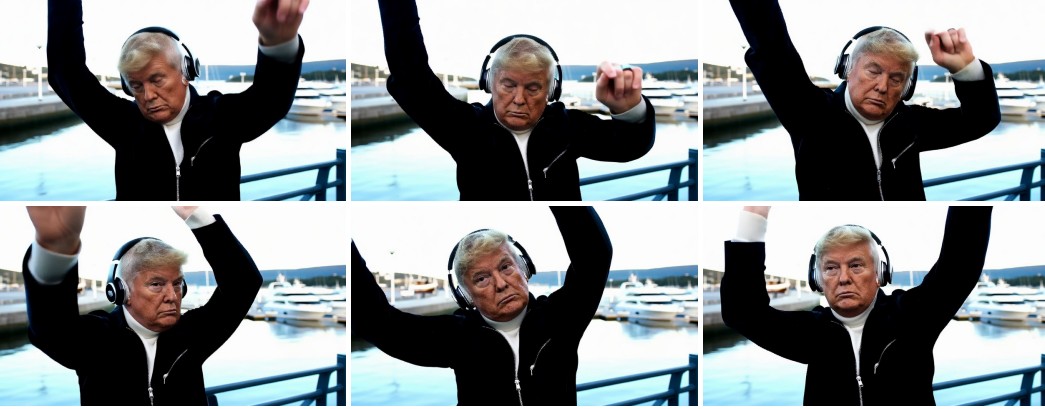

*The video features a person wearing headphones and a black outfit, standing on a rooftop terrace with a panoramic view of the city skyline... The person is seen with their arms raised above their head, appearing to be dancing or moving rhythmically to music...*

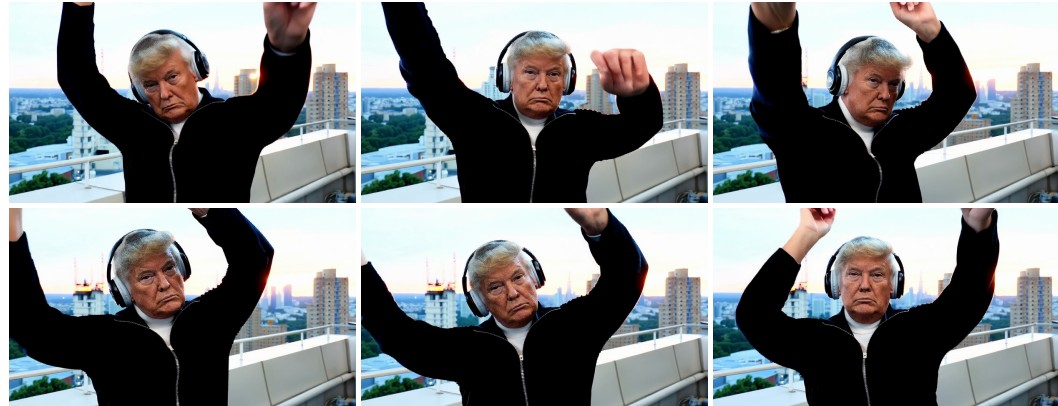

Figure 18: More showcases of identity-preserving videos generated by our FaithfulFaces.

Input Reference Image and Text Prompt 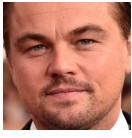
*The video features a person in an early morning urban street setting, wearing a light gray long-sleeve shirt and black boxing gloves. The individual is engaged in a boxing routine, demonstrating various punches and defensive maneuvers...*

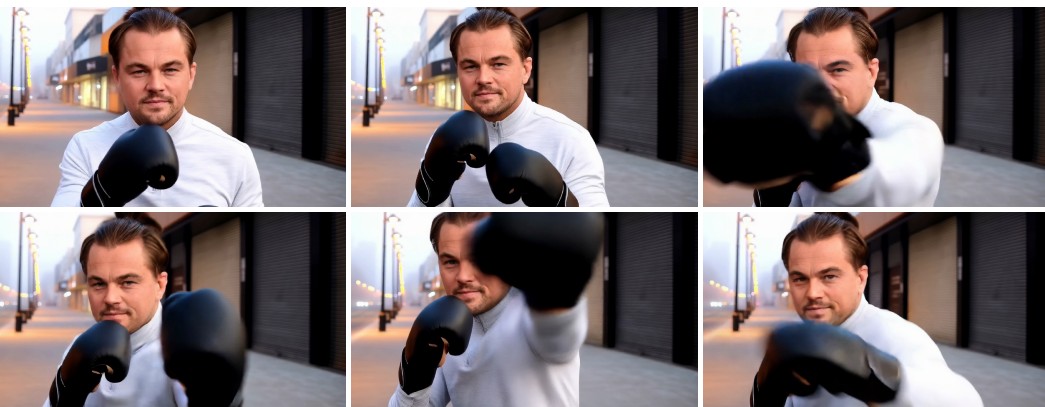

*The video features a ballerina performing a graceful dance move in a dimly lit studio... The ballerina is wearing a black and white leotard with long sleeves that extend to her wrists, and her hair is neatly tied back in a bun...*

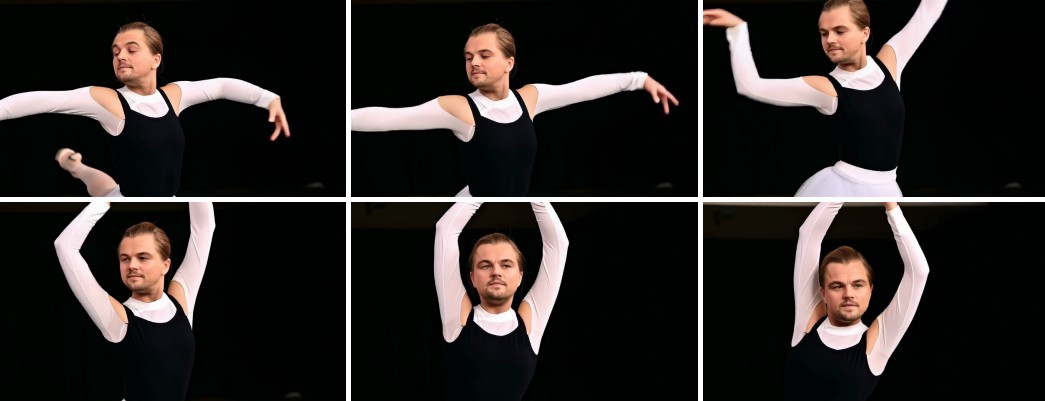

*The video is shot inside a dimly lit city subway platform at night... The person, wearing a black jacket and dark jeans, stands near the platform edge with a blurred train speeding by in the background. At the beginning, the person has their back to the camera and slowly turns to face it...*

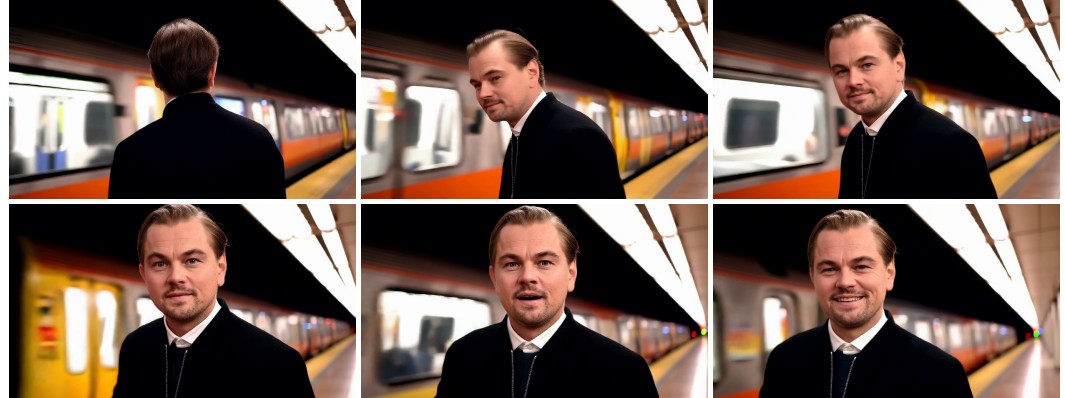

Figure 19: More showcases of identity-preserving videos generated by our FaithfulFaces.

Input Reference Image and Text Prompt 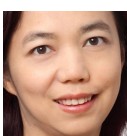 *The video is shot inside a dimly lit city subway platform at night... The person, wearing a black jacket and dark jeans, stands near the platform edge with a blurred train speeding by in the background. At the beginning, the person has their back to the camera and slowly turns to face it...*

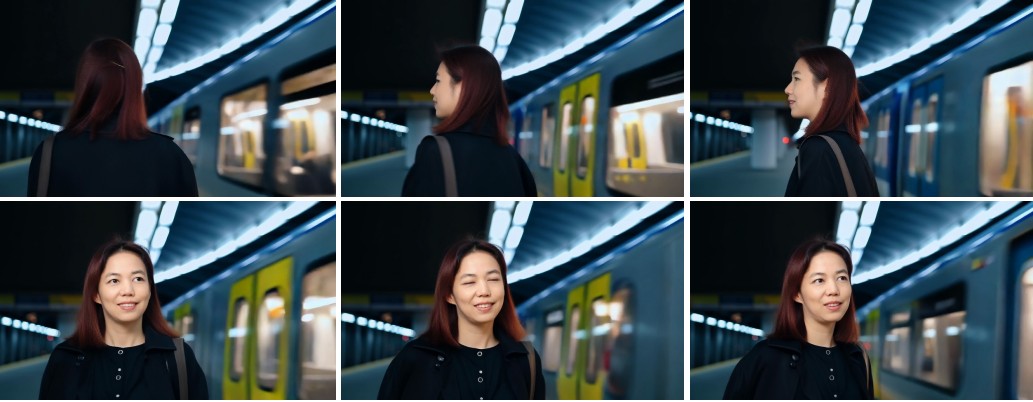

*The video features a person wearing headphones and a black outfit, standing outdoors in an urban setting... The person is seen with their arms raised above their head, appearing to be dancing or moving rhythmically to music...*

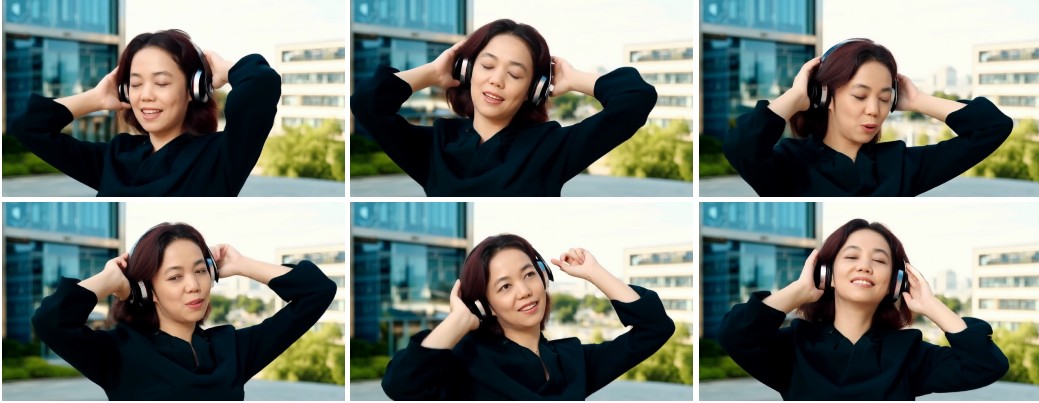

*The video features a person working out in a cozy living room, wearing a light gray long-sleeve shirt and black boxing gloves. The individual is engaged in a boxing routine, demonstrating various punches and defensive maneuvers...*

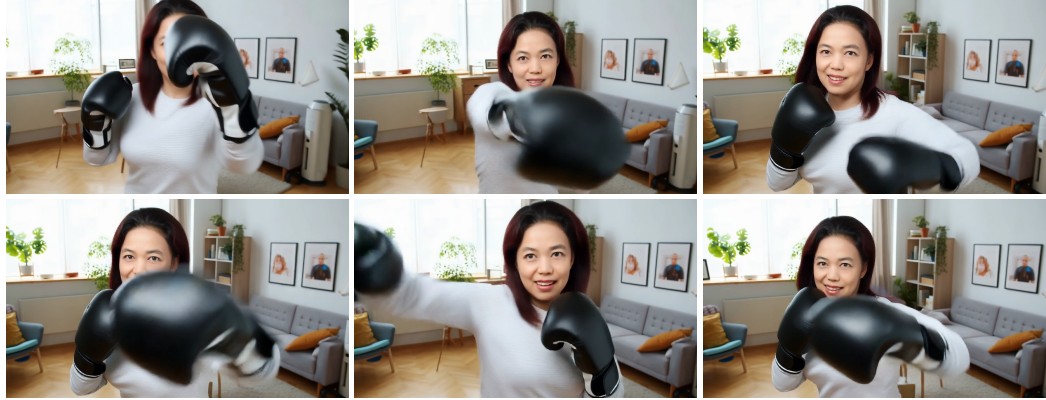

Figure 20: More showcases of identity-preserving videos generated by our FaithfulFaces.

