# OpenReview forum: "FaithfulFaces: Pose-Faithful Facial Identity Preservation for Text-to-Video Generation"
_ICLR.cc/2026/Conference — Submitted to ICLR 2026_

### Official Review · Reviewer_9GSU · 2025-10-25

**Soundness:** 3
**Presentation:** 3
**Contribution:** 2
**Rating:** 4
**Confidence:** 4

**Summary:**

The paper proposes a pose-shared identity aligner that learns a dictionary-based global facial-pose representation from a single reference image, optimized via a pose variation–identity invariance contrastive loss and explicit Euler-angle embeddings.  A data pipeline selects single-subject videos with strong pose variation and produces descriptive prompts.  Across diverse prompts and identities, the method improves FaceSim and FID while maintaining text alignment, outperforming open-source and commercial baselines on qualitative and quantitative evaluations.

**Strengths:**

1. A pose-shared dictionary with Euler-angle embeddings gives the model an explicit global head-pose prior from a single image, reducing identity drift during motion.

2. The pose variation–identity invariance (InfoNCE) loss aligns poses of the same identity and discourages collapse, supported by an information-theoretic view and t-SNE plots.

3. A pipeline filters single-subject videos by measured pose variation and auto-generates prompts, yielding 51,624 samples focused on difficult pose dynamics.

4. FaithfulFaces delivers best FaceSim-Cur/Arc, lower FID, and solid CLIPScore across 600 test videos, surpassing open-source and commercial baselines.

**Weaknesses:**

1. The pipeline and aligner rely on 6DRepNet head-pose estimates and a fixed variation threshold (120); mis-estimation or thresholding bias could skew data and priors.

2. Training uses VACE-14B with LoRA on 32 H20 GPUs. Reproducibility for typical labs may be limited without ablations on smaller backbones or ranks.

3. Some baselines are commercial black boxes and others use different foundations (Wan, CogVideoX, Hunyuan), complicating strictly like-for-like attribution.

4. The method targets single-subject, face-centric videos. Multi-person scenes, rapid occluders, or non-frontal identity cues (gait, hair) are not deeply evaluated.

**Questions:**

1. The robustness of aligner when the pose estimator is noisy or biased across demographics should be validated.

2. Can the dictionary be adapted online to a new identity with few frames without retraining the generator?

3. How does performance scale on lighter backbones or lower-rank LoRA, and what is the compute–quality trade-off?

4. The extensibility of method  to multi-subject scenes with identity disambiguation and cross-shot continuity should be added.

---

> ### Author Response · Authors · 2025-11-20
> **Thank you for recognizing our contributions and promising results. Below, we respond to each Question (Q) with an Answer (A).**
>
> ***Part 1 / 2***
>
> **Q1:** The pipeline and aligner rely on 6DRepNet head-pose estimates and a fixed variation threshold (120); mis-estimation or thresholding bias could skew data and priors.
>
> **A1:** Thanks for your comment. For your concerns, we make the clarifications below:
>
> i) We fully agree with your points regarding pose estimation accuracy and threshold setting.  Precisely for this reason, we implemented two steps during dataset construction to mitigate these issues as much as possible: 1) The threshold of 120 was determined by manually annotating 2,000 videos, not through any automated method, ensuring sufficient reliability (please see lines 338-343 of our submission). 2) After automated processing, we conducted rigorous manual verification for qualification (please see lines 351-353 of our submission). These procedures are critical steps for training a reliable aligner, as unreliable data or thresholds would have made it impossible to achieve good results.
>
> ii) The sparsity design of the dictionary representation mechanism in our aligner inherently possesses a certain degree of noise tolerance. To demonstrate this, we conducted additional ablation experiments involving Euler angle perturbations. Specifically, given four perturbation ranges for Euler angles ($-5^\circ \sim +5^\circ$, $-10^\circ \sim +10^\circ$, $-15^\circ \sim +15^\circ$, $-20^\circ \sim +20^\circ$), we applied random perturbations within these ranges to the predicted Euler angles to observe the impact on performance. The experimental results are shown in the table below, we can observe that performance exhibits only minor variations within the perturbation range of $-15^\circ \sim +15^\circ$. Significant performance degradation occurs only when perturbations exceed $15^\circ$ (representing substantial errors). These results prove that our method exhibits high robustness to Euler angle errors within a certain range.
>
> | Perturbation Range         | FaceSim-Cur | FaceSim-Arc | FID    | CLIPScore |
> | :- | :---------- | :---- | :----- | :-------- |
> | No Perturbation       | 0.568       | 0.542       | 164.24 | 33.93     |
> | $-5^\circ \sim +5^\circ$   | 0.566       | 0.540       | 164.54 | 33.89     |
> | $-10^\circ \sim +10^\circ$ | 0.557       | 0.531       | 166.06 | 33.77     |
> | $-15^\circ \sim +15^\circ$ | 0.552       | 0.526       | 167.07 | 33.74     |
> | $-20^\circ \sim +20^\circ$ | 0.523       | 0.499       | 172.54 | 33.56     |
>
> **Q2:** Training uses VACE-14B with LoRA on 32 H20 GPUs. Reproducibility for typical labs may be limited without ablations on smaller backbones or ranks.
>
> **A2:** Thanks for your comment. For your concerns, we make the clarifications below:
>
> i) The VACE-14B model, which is based on Wan-14B, was used to ensure high quality and facilitate a fair comparison, as most Wan-based comparative algorithms utilize the 14B model for their experiments (e.g., SkyReels-A2, MAGREF, Stand-In, Phantom). This ensures that our proposed innovations have practical application value and the high performance ceiling.
>
> ii) Our method can seamlessly adapt to smaller models. To address your concerns, we conducted an additional ablation study by replacing the base model VACE-14B with the lightweight VACE-1.3B. The experimental results are shown in the table below, demonstrating that our method still yields significant performance improvements even on a smaller backbone network.
>
> | Methods  | FaceSim-Cur | FaceSim-Arc | FID   | CLIPScore |
> | :- | :---------- | :-- | :--------- | :-------- |
> | Baseline-1.3B      | 0.247       | 0.232       | 203.76     | 31.69     |
> | FaithfulFaces-1.3B | **0.413**   | **0.395**   | **193.68** | **33.37** |
>
> **Q3:** Some baselines are commercial black boxes and others use different foundations (Wan, CogVideoX, Hunyuan), complicating strictly like-for-like attribution.
>
> **A3:** We are deeply grateful to you for raising this insightful question. We fully understand and acknowledge that a rigorous 'like-for-like' comparison is indeed an inherent challenge in the rapidly evolving field of generative AI, especially when dealing with commercial black-box models.
>
> Since our method is based on Wan, the most popular and powerful open-source model, we can isolate and compare the Wan-based methods separately. Furthermore, to further enhance the comprehensiveness of the comparison, we additionally included three advanced Wan-based methods: Concat-ID-Wan, SkyReels-A2, and MAGREF. The quantitative results are shown in the table below. We can observe that our method achieves the best performance among Wan-based approaches.
>
> |Methods|FaceSim-Cur|FaceSim-Arc|FID| CLIPScore |
> |:-|:-|:-|:-|:-|
> |VACE-14B|0.403|0.382|191.02|31.83|
> |Stand-In  | 0.415| 0.395   | 196.21| 30.38|
> |Phantom-14B| 0.484   | 0.456 | 214.99| 29.67 |
> |Concat-ID-Wan| 0.408   | 0.387  | 189.55| 31.49|
> |SkyReels-A2| 0.410| 0.384| 237.29 | 28.10|
> |MAGREF| 0.417 | 0.392| 207.69 | 31.13  |
> |FaithfulFaces (Ours)|**0.568**|**0.542**|**164.24**|**33.93**|

---

> > ### Author Response · Authors · 2025-11-20
> >
> > ***Part 2 / 2***
> >
> > **Q4:** The method targets single-subject, face-centric videos. Multi-person scenes, rapid occluders, or non-frontal identity cues (gait, hair) are not deeply evaluated.
> >
> > **A4:** Thanks for your comment. For your concerns, we make the clarifications below:
> >
> > i) Firstly, it is necessary to clarify that our method focuses only on single-identity consistent generation, which is explicitly stated in both the method description and the dataset construction. Secondly, we are currently dedicated to extending our method to multi-identity scenarios. The proposed aligner can indeed be applied to multi-identity settings, as it can independently encode the corresponding global representation for each identity. However, multi-identity scenarios present task-specific challenges, such as identity mixing. This requires further exploration and resolution, but it does not diminish the contribution of this paper, as this constitutes future research work.
> >
> > ii) For the IPT2V task, users typically provide frontal face images as input, as frontal faces offer comprehensive facial structure information and facial texture details, whereas non-frontal faces often lack sufficient texture and structural details necessary for high-fidelity identity preservation. Consequently, using non-frontal faces as input inevitably leads to degraded performance.
> >
> > iii) Empirically, our proposed pose-shared identity aligner can mitigate the performance degradation when using non-frontal faces. This is because our aligner can encode global representations from an input face image. To demonstrate this, we collected 10 IDs with both frontal and non-frontal face images for ablation and comparative (strongest baseline Phantom as the representative) experiments. The results are shown in the table below, where the values reported in each cell denote FaceSim-Cur/ FaceSim-Arc. We observe that both our model without the aligner and the strongest baseline Phantom suffer a severe performance decrease exceeding 50% when non-frontal face images are used as input. In contrast, our model with the aligner is able to control the performance decrease within 25%. These results provide strong evidence that the pose-shared identity aligner can mitigate performance degradation in non-frontal face scenarios.
> >
> > | Methods                      | Frontal     | Non-frontal    |
> > | :--------------------------- | :---------- | :------------------------------------------------------ |
> > | Phantom (Strongest baseline) | 0.470/0.435 | 0.231 ($\downarrow$ 50.85%)/0.216 ($\downarrow$ 50.34%) |
> > | Ours w/o aligner             | 0.448/0.423 | 0.202 ($\downarrow$ 54.91%)/0.197 ($\downarrow$ 53.43%) |
> > | Ours                         | 0.544/0.522 | 0.409 ($\downarrow$ 24.82%)/0.392 ($\downarrow$ 24.90%) |
> >
> > **Q5:** Can the dictionary be adapted online to a new identity with few frames without retraining the generator?
> >
> > **A5:** We thank you for this insightful question. Our current framework prioritizes a strong zero-shot generalization capability via the general dictionary. We acknowledge that your proposed online adaptation mode is valuable for specific needs. Conceptually, a direct approach could involve adapting the dictionary with lightweight LoRA for rapid adaptation to new identities. We view this as a promising direction for future efficiency-focused extensions.
> >
> > We sincerely thank you once again for allocating your valuable time to provide such profound, pertinent, and highly constructive feedback on our work. We have executed detailed and targeted responses and revisions based on your guidance. We genuinely hope that these comprehensive responses and modifications will fully address your concerns and lead you to grant a higher recognition and evaluation of our work.

---

> > > ### Comment · Reviewer_9GSU · 2025-11-24
> > >
> > > Thanks for your response. My concerns have been resolved.

---

> > > > ### Author Response · Authors · 2025-11-24
> > > > **Thank you for acknowledging our response and raising the score**
> > > >
> > > > Thank you for your response. We are glad to hear that your concerns have been satisfactorily addressed. We appreciate your time in re-evaluating our work and increasing the score.

---

### Official Review · Reviewer_mLYU · 2025-10-28

**Soundness:** 3
**Presentation:** 2
**Contribution:** 3
**Rating:** 4
**Confidence:** 3

**Summary:**

This paper introduces FaithfulFaces, a novel approach designed to overcome the limitations of existing identity-preserving Text-to-Video (T2V) methods, particularly when facing large facial pose variations or occlusions. These challenging scenarios often lead to identity distortion in generated videos. The core contribution is the pose-shared identity aligner module. The pose-shared identity aligner is engineered to extract robust, pose-disentangled identity features from an identity image and spatially align them with the pose information derived from the driving image. This spatially-aligned identity feature is then injected into multiple layers of the T2V U-Net architecture via a spatial modulation mechanism. Experimental results demonstrate that FaithfulFaces significantly improves identity consistency, especially under extreme pose changes, outperforming current state-of-the-art techniques.

**Strengths:**

1. The pose-shared identity aligner is a clever and effective design. By pre-aligning the identity features based on the shared pose context, it effectively mitigates the issue of identity feature misalignment under large pose transformations, which is a major weakness in many identity-embedding approaches.
2.  The quantitative (especially ID preservation metrics) and qualitative results strongly validate the method's superiority in large pose cases.
3. The method is modularly designed as a plug-in component, allowing for straightforward integration into prevalent T2V diffusion model architectures (e.g., U-Net), which enhances its practical applicability and adoption potential.

**Weaknesses:**

1. The introduction of the pose-shared identity aligner module and multiple spatial modulation layers inevitably increases the model's parameter count and inference latency. The paper currently lacks a detailed comparison of inference speed, memory consumption (VRAM), and model size relative to the baseline T2V model. This information is vital for real-world deployment.
2. Although the generated videos appear smooth qualitatively, the paper does not include specific quantitative metrics for video temporal consistency (e.g., using metrics like LPIPS on consecutive frames or Frame Coherence metrics). Temporal stability is crucial, especially when large, frame-by-frame pose changes are involved.
3. The entire focus is on facial identity. While successful, the PFA provides no direct benefit or conditioning for non-face elements (e.g., clothes, background, scene style). This is an inherent limitation of the identity-specific approach, though not a flaw in the paper's core claim.
4. The current ablation studies are insufficient to conclusively determine the contribution of the pose-shared identity aligner. Further ablation studies are required, specifically by replacing the feature used (e.g., substituting the current feature with an ArcFace feature or another comparable representation) to isolate and measure the aligner's specific effect.

**Questions:**

1. The success of pose-shared identity aligner relies on extracting a pure, pose-disentangled identity feature. How robust is the identity extraction process when the reference image itself is highly non-frontal (e.g., a strong side profile), partially occluded, or of poor quality? Could the authors provide a comparative result showing the ID consistency metrics (e.g., ID loss) when the reference image is deliberately a non-frontal view? This would better validate.
2. I am confused regarding the feature interaction within the PIA. The process appears to involve independent feature reconstruction based on a shared dictionary, rather than a direct interaction between the two image features. While a shared dictionary implies some identity-related commonality, the mechanism by which this leads to a truly pose-invariant identity feature is not fully clear.


Minor
1.  Pose-shared Facial Aligner in Figure 2 and  pose-shared identity aligner in Figure 3. I suppose there are the same thing?

---

> ### Author Response · Authors · 2025-11-20
> **Thank you for recognizing our contributions and promising results. Below, we respond to each Question (Q) with an Answer (A).**
>
> ***Part 1 / 2***
>
> **Q1:** The introduction of the pose-shared identity aligner module and multiple spatial modulation layers inevitably increases the model's parameter count and inference latency. The paper currently lacks a detailed comparison of inference speed, memory consumption (VRAM), and model size relative to the baseline T2V model. This information is vital for real-world deployment.
>
> **A1:** Thanks for your comment. The model size of the pose-shared identity aligner and the LoRA module is very lightweight compared to the baseline T2V model, resulting in minimal increases in inference speed and memory consumption. To demonstrate this, we first list the parameter counts for the Baseline Model, the Pose-shared Identity Aligner, and the LoRA Module in the table below. It can be intuitively concluded that the parameter counts of the Aligner (0.252B) and LoRA (0.103B) are marginal compared to the Baseline Model (14B).
>
> |Baseline Model|Pose-shared Identity Aligner|LoRA Module|
> |:-|:-|:-|
> |14B|0.252B|0.103B|
>
> We further provide a comparison between the Baseline Model and our method regarding inference speed and memory consumption, with the quantitative results listed in the table below. Quantitatively, the increase in inference speed and memory consumption brought by these lightweight modules is marginal.
>
> |Metrics|Baseline|Ours|
> |:-|:-|:-|
> |Inference Speed|127s|129s ($\uparrow$ 1.57%)|
> |Memory Consumption (VRAM)|70339MiB|71833MiB ($\uparrow$ 2.12%)|
>
> **Q2:** Although the generated videos appear smooth qualitatively, the paper does not include specific quantitative metrics for video temporal consistency (e.g., using metrics like LPIPS on consecutive frames or Frame Coherence metrics). Temporal stability is crucial, especially when large, frame-by-frame pose changes are involved.
>
> **A2:** Thanks for your comment. For your concerns, we make the clarifications below:
>
> i) The evaluation metrics used in our work follow prior works (e.g., ConsisID), primarily focusing on identity consistency. Furthermore, benefiting from the powerful capabilities of modern foundational T2V models, our method naturally maintains high temporal consistency.
>
> ii) To further address your concerns, we provide the quantitative results of different methods on the most commonly used Frame Coherence metrics (CLIP-based adjacent frame similarity) in the table below. We observe that all methods exhibit high temporal consistency, with minimal variation in temporal consistency between different methods. However, there are considerable variations in identity consistency across different methods.
>
> |Methods|Frame Coherence|FaceSim-Cur|FaceSim-Arc|
> |:-|:-|:-|:-|
> |Vidu|0.950|0.293|0.278|
> |Kling|0.979|0.447|0.416|
> |ConsisID|0.957|0.365|0.350|
> |VACE-14B|0.972|0.403|0.382|
> |HunyuanCustom|0.979|0.453|0.432|
> |Stand-In|0.978|0.415|0.395|
> |Phantom-14B|0.978|0.484|0.456|
> |Concat-ID-Wan|0.977|0.408|0.387|
> |SkyReels-A2|0.970|0.410|0.384|
> |MAGREF|0.974|0.417|0.392|
> |FaithfulFaces (Ours)|0.978|0.568|0.542|
>
> **Q3:** The entire focus is on facial identity. While successful, the PFA provides no direct benefit or conditioning for non-face elements (e.g., clothes, background, scene style). This is an inherent limitation of the identity-specific approach, though not a flaw in the paper's core claim.
>
> **A3:** Thanks for your comment. In the IPT2V field, the research objective is focused on identity consistency, because foundational T2V models cannot perfectly generate consistent identities, which is the reason why many works/commercial models are dedicated to solving this problem. Correspondingly, our work also focuses on the issue of identity consistency. Non-face elements are beyond the scope of IPT2V field including our work. Of course, your comment is also very insightful and worth exploring in future research. Some of the technical strategies of our method may have the potential to be extended to non-face scenarios.
>
> **Q4:** The current ablation studies are insufficient to conclusively determine the contribution of the pose-shared identity aligner. Further ablation studies are required, specifically by replacing the feature used (e.g., substituting the current feature with an ArcFace feature or another comparable representation) to isolate and measure the aligner's specific effect.
>
> **A4:** We thank you for this constructive suggestion. According to your suggestion, we conducted additional ablation studies by replacing the aligner features with ArcFace features. The experimental results are shown in the table below. We can observe that performance significantly deteriorates when using ArcFace features. The underlying reason is that the ArcFace feature is unable to represent global facial pose information. This ablation study further demonstrates the effectiveness of the pose-shared identity aligner.
> |Feature Type|FaceSim-Cur|FaceSim-Arc|FID|CLIPScore|
> |:-|:-|:-|:-|:-|
> |ArcFace|0.475|0.453|177.20|32.70|
> |Aligner (Ours)|**0.568**|**0.542**|**164.24**|**33.93**|

---

> > ### Author Response · Authors · 2025-11-20
> >
> > ***Part 2 / 2***
> >
> > **Q5:** The success of the pose-shared identity aligner relies on extracting a pure, pose-disentangled identity feature. How robust is the identity extraction process when the reference image itself is highly non-frontal (e.g., a strong side profile), partially occluded, or of poor quality? Could the authors provide a comparative result showing the ID consistency metrics (e.g., ID loss) when the reference image is deliberately a non-frontal view? This would better validate.
> >
> > **A5:** Thanks for your comment. For your concerns, we make the clarifications below:
> >
> > i) For the IPT2V task, users typically provide frontal face images as input, as frontal faces offer comprehensive facial structure information and facial texture details, whereas non-frontal faces often lack sufficient texture and structural details necessary for high-fidelity identity preservation. Consequently, using non-frontal faces as input inevitably leads to degraded performance.
> >
> > ii) Empirically, our proposed pose-shared identity aligner can mitigate the performance degradation when using non-frontal faces. This is because our aligner can encode global representations from an input face image. To demonstrate this, we collected 10 IDs with both frontal and non-frontal face images for ablation and comparative (strongest baseline Phantom as the representative) experiments. The results are shown in the table below, where the values reported in each cell denote FaceSim-Cur/ FaceSim-Arc. We observe that both our model without the aligner and the strongest baseline Phantom suffer a severe performance decrease exceeding 50% when non-frontal face images are used as input. In contrast, our model with the aligner is able to control the performance decrease within 25%. These results provide strong evidence that the pose-shared identity aligner can mitigate performance degradation in non-frontal face scenarios.
> >
> > | Methods                      | Frontal     | Non-frontal                                             |
> > | :--------------------------- | :---------- | :------------------------------------------------------ |
> > | Phantom (Strongest baseline) | 0.470/0.435 | 0.231 ($\downarrow$ 50.85%)/0.216 ($\downarrow$ 50.34%) |
> > | Ours w/o aligner             | 0.448/0.423 | 0.202 ($\downarrow$ 54.91%)/0.197 ($\downarrow$ 53.43%) |
> > | Ours                         | 0.544/0.522 | 0.409 ($\downarrow$ 24.82%)/0.392 ($\downarrow$ 24.90%) |
> >
> > **Q6:** I am confused regarding the feature interaction within the PIA. The process appears to involve independent feature reconstruction based on a shared dictionary, rather than a direct interaction between the two image features. While a shared dictionary implies some identity-related commonality, the mechanism by which this leads to a truly pose-invariant identity feature is not fully clear.
> >
> > **A6:** Thanks for your comment. For your concerns, we make the clarifications below:
> >
> > i) In the PIA, different face features are represented via a shared dictionary and are constrained by the contrastive loss to align in a global and pose-invariant space. Therefore, the entire process is not strictly independent.
> >
> > ii) The visualization of the encoded facial identity shown in Figure 6 proves that PIA is able to map faces of the same identity but with different facial poses into a shared feature space. Furthermore, Figure 7 reveals some meaningful activation patterns for the learned dictionary, wherein images with similar poses tend to frequently activate particular dictionary elements. These results collectively demonstrate that PIA can encode reliable pose-invariant identity features.
> >
> > **Q7:** Pose-shared Facial Aligner in Figure 2 and pose-shared identity aligner in Figure 3. I suppose there are the same thing?
> >
> > **A7:** We thank you for pointing out this inconsistency in terminology between Figure 2 and Figure 3. We confirm that the Pose-shared Facial Aligner (Figure 2) and the Pose-shared Identity Aligner (Figure 3) refer to the exact same module. We apologize for this potential confusion. In our revised submission, we have modified the Pose-shared Facial Aligner in Figure 2 to Pose-shared Identity Aligner.
> >
> > We sincerely thank you once again for allocating your valuable time to provide such profound, pertinent, and highly constructive feedback on our work. We have executed detailed and targeted responses and revisions based on your guidance. We genuinely hope that these comprehensive responses and modifications will fully address your concerns and lead you to grant a higher recognition and evaluation of our work.

---

> ### Comment · Reviewer_mLYU · 2025-11-26
> **New concern**
>
> Thank you for your reply. Since this is a diffusion framework, even though the inference time and parameters increased slightly, it is still acceptable. Regarding temporal consistency, I checked the demos again and referred to the metrics the authors provided. It is convincing to me.
>
> While I acknowledge the improved facial fidelity and temporal consistency, I have noticed that the aggressive optimization for facial identity appears to compromise the generation quality of non-face regions. For instance, in Figure 14 (second case), the subject's hands appear unnatural and distorted. Since your method introduces a specific constraint ($\mathcal{L}_{PIA}$) and module focused entirely on the face, is there a trade-off where the model 'forgets' or degrades its prior knowledge for general human anatomy (like hands and limbs)? Does the Identity Aligner disrupt the semantic consistency between the head and the rest of the body? Also, you can check that some regions are strange in demo videos, like the hand in fig14_2 demo.
>
> Regarding A5, while the quantitative comparison demonstrates that your method suffers less degradation (24.82%) than the baseline (50.85%) when using non-frontal inputs, the absolute performance gap between frontal (0.544) and non-frontal (0.409) inputs remains significant. Furthermore, could the authors provide qualitative visual comparisons for these non-frontal reference cases in the rebuttal PDF? Specifically, when the reference is a side profile (lacking half the facial texture), how does the model hallucinate the occluded side? Does it maintain faithful identity, or does it revert to a generic average face that satisfies the pose dictionary? Visual evidence is crucial here to validate that the higher FaceSim score isn't just detecting a generic high-quality face.

---

> > ### Author Response · Authors · 2025-11-27
> >
> > Thank you very much for your response. We are pleased that our rebuttal has addressed most of your concerns.
> >
> > Regarding your further concerns and suggestions, we would like to make the following clarifications:
> >
> > **1)** We would like to clarify that our method employs a LoRA (Low-Rank Adaptation) fine-tuning strategy. During training, the parameters of the pre-trained text-to-video foundation model are completely frozen. We only update the parameters of the additional LoRA layers and the specific identity modules.  Therefore, our method does not "forget" or degrade its prior knowledge of general human anatomy, nor does it disrupt the semantic consistency inherent in the base model. The foundation model's original capabilities are preserved.
> >
> > **2)** Regarding the issue of unnatural hands you mentioned, we first want to sincerely thank you for your careful examination. We also acknowledge that some hand unnaturalness indeed exists in the second case of Figure 14. However, we need to clarify that this unnaturalness is primarily attributed to the inherent limitations of the baseline model, and it is also widely present in existing SOTA methods.
> >
> > To demonstrate this, we have provided visual evidence in Figure 12 of the revised submission, using the second case from Figure 14 (which you mentioned) as an example. In this visualization, severe hand distortion and collapse are marked with red boxes. We can observe that unnaturalness, distortion, and even collapse in the hand region are widespread in existing methods. Therefore, this is not caused by our identity constraint or identity aligner. In fact, the distortion and collapse of hands is an open issue within text-to-video foundational models. Empirically, we could construct "good-bad" pairs and use Direct Preference Optimization to encourage the model to reduce the unnaturalness of the hands. However, this genuinely exceeds the scope of the IPT2V field. Of course, we also believe that these issues will be gradually addressed through the continuous progress of the entire video generation community.
> >
> > **3)** We are very grateful for your further suggestions. We fully agree that providing visual results of non-frontal reference cases would further demonstrate the effectiveness of our method. As requested, we have provided qualitative comparisons in Figure 11 of the revised submission. Specifically, in Figure 11, we have provided visualization results for both frontal-view and non-frontal view reference cases of the same identity. We can observe that when using a non-frontal image as input, the faces generated by the SOTA method Phantom and Ours w/o Aligner completely collapse. In contrast, our method still maintains identity consistency. These visual results once again demonstrate that our method is capable of improving identity consistency in non-frontal face scenarios.
> >
> > Furthermore, we need to reiterate that the performance gap between frontal and non-frontal inputs is empirically expected. This stems from the inherent information loss in side-profile images, where nearly half of the facial texture and key identity features are obscured. Maintaining identity features from such a local view is considered an ill-posed problem, and consequently, the similarity score is necessarily lower compared to a fully visible frontal reference image. At the same time, we believe these additional visual results fundamentally reinforce the contributions of our work.
> >
> > Thank you again for your valuable comments and suggestions. We hope that these further clarifications and revisions fully address your concerns.

---

### Official Review · Reviewer_c9jn · 2025-10-30

**Soundness:** 3
**Presentation:** 3
**Contribution:** 3
**Rating:** 4
**Confidence:** 4

**Summary:**

The paper proposes FaithfulFaces to address identity preservation in complex dynamic scenes. It introduces a pose-shared identity aligner and a dedicated video dataset pipeline with substantial facial pose variations. Experiments show that FaithfulFaces achieves state-of-the-art performance, generating high-quality videos with clear facial structures and consistent identity preservation.

**Strengths:**

(1) FaithfulFaces is compared with the closed-source methods Vidu and Kling, demonstrating the superiority of the proposed approach.

(2) The effectiveness of the method is validated through both quantitative and qualitative experiments, and the contributions of each module are analyzed via ablation studies.

**Weaknesses:**

The paper lacks citations, comparisons, or discussions of prior representative work such as Concat-ID [1],SkyReels-A2 [2], and MAGREF [3] (all based on Wan and open-source), indicating that the authors may not be very familiar with the field of identity-preserving video generation.

[1] Concat-ID: Towards Universal Identity-Preserving Video Synthesis

[2] SkyReels-A2: Compose Anything in Video Diffusion Transformers

[3] MAGREF: Masked Guidance for Any-Reference Video Generation with Subject Disentanglement

**Questions:**

My recommendation could improve if the authors provide convincing responses to the points raised.

---

> ### Author Response · Authors · 2025-11-20
> **Thank you for recognizing our contributions and promising results. Below, we respond to each Question (Q) with an Answer (A).**
>
> **Q1:** The paper lacks citations, comparisons, or discussions of prior representative work such as Concat-ID \[1], SkyReels-A2 \[2], and MAGREF \[3] (all based on Wan and open-source), indicating that the authors may not be very familiar with the field of identity-preserving video generation.
>
> **A1:** We thank you for recommending Concat-ID, SkyReels-A2, and MAGREF. We agree that comparing against these specific Wan-based extensions provides a more comprehensive evaluation of our method within this ecosystem.
>
> For Concat-ID and SkyReels-A2, they were actually already discussed and compared in our Wan-based comparison method Stand-In. Since we focused on a comparison involving a broader range of architectures, certain trade-offs were made in the comparison. MAGREF was posted to the arXiv platform on October 10, 2025, which was after the paper submission deadline of the ICLR 2026 (September 24, 2025). However, we fully agree that explicit comparisons with more recent Wan-based methods strengthen the paper.
>
> Following your suggestion, we conducted comparisons with the aforementioned methods, and the experimental results are shown in the table below. We can observe that our proposed method still outperforms these approaches. The complete comparison results can be found in Table 1 of the revised submission. Furthermore, we have added citations and discussions of these three works in the Related Work section.
>
> | Methods             | FaceSim-Cur | FaceSim-Arc | FID        | CLIPScore |
> | :------------------ | :---------- | :---------- | :--------- | :-------- |
> | Concat-ID-Wan       | 0.408       | 0.387       | 189.55     | 31.49     |
> | SkyReels-A2         | 0.410       | 0.384       | 237.29     | 28.10     |
> | MAGREF              | 0.417       | 0.392       | 207.69     | 31.13     |
> | FaithfulFaces (Ours) | **0.568**   | **0.542**   | **164.24** | **33.93** |
>
> We sincerely thank you once again for allocating your valuable time to provide such profound, pertinent, and highly constructive feedback on our work. We have executed detailed and targeted responses and revisions based on your guidance. We genuinely hope that these comprehensive responses and modifications will fully address your concerns and lead you to grant a higher recognition and evaluation of our work.

---

> > ### Comment · Reviewer_c9jn · 2025-11-22
> >
> > We appreciate your response and the effort you put into the related experiments. However, I am still somewhat unclear about the experiment.
> >
> > MAGREF: Masked Guidance for Any-Reference Video Generation with Subject Disentanglement (https://arxiv.org/abs/2505.23742
> > ) had its paper publicly released in May and its code released in June, as confirmed by the arXiv identifier (2505.23742) and the GitHub code submission date (https://github.com/MAGREF-Video/MAGREF/
> > ). The authors, however, stated that MAGREF was posted to arXiv on October 10, 2025, which was after the paper submission deadline of the ICLR 2026 (September 24, 2025). We kindly request that the authors verify whether the code used corresponds to the correct paper.

---

> > > ### Author Response · Authors · 2025-11-22
> > >
> > > Thank you very much for your prompt response and for pushing us to be more rigorous.
> > >
> > > Regarding MAGREF, the version of the paper we referred to was the latest one dated October 10 (v2, October, https://arxiv.org/pdf/2505.23742). Based on the link you provided, we failed to notice that there was an older version of this work from May (v1, May, https://arxiv.org/pdf/2505.23742v1).
> > >
> > > Regarding the experiment and code verification. We have immediately re-examined the papers and the codebase. We confirm that two arXiv versions point to the same official GitHub repository (https://github.com/MAGREF-Video/MAGREF/). Therefore, we are certain that the code we used is correct and is the latest version. We believe that using the latest version for comparison is more reliable.
> > >
> > > Thank you again for your careful observation, guidance, and comments on our paper. We hope this response can address your concerns completely.

---

> > > > ### Comment · Reviewer_c9jn · 2025-11-22
> > > >
> > > > Thank you for your detailed and thoughtful reply.
> > > >
> > > > I would like to clarify that in my initial comment, I did not provide any link, but only cited the work by its reference. Therefore, I am unsure how to interpret the statement: “Based on the link you provided, we failed to notice that there was an older version of this work from May.”
> > > >
> > > > Thank you again for your clarification. I will take your explanation, along with the feedback from the other reviewers, into account when making my decision regarding the score.

---

> > > > > ### Author Response · Authors · 2025-11-22
> > > > >
> > > > > Thank you for your further response.
> > > > >
> > > > > Regarding your concerns about our statement, we provide further clarifications below:
> > > > > 1. The phrase 'Based on the link you provided' refers to the link included in your previous reply (i.e., https://arxiv.org/abs/2505.23742).
> > > > > 2. This statement means that after reviewing this link, we realized we had overlooked an earlier preprint version of MAGREF. Therefore, we immediately reviewed the two preprint versions based on your suggestion. We are sorry for any confusion caused by our statement.
> > > > >
> > > > > Thank you again for your prompt response. We hope these further clarifications will fully address your concerns.

---

### Official Review · Reviewer_qQ3h · 2025-11-03

**Soundness:** 3
**Presentation:** 2
**Contribution:** 2
**Rating:** 6
**Confidence:** 4

**Summary:**

This paper introduces FaithfulFaces, a pose-faithful facial identity preservation learning framework for the identity-preserving text-to-video (IPT2V) task. The method centers on a pose-shared identity aligner that refines and encodes global facial pose representations from a single reference image using a pose-shared dictionary and a pose variation-identity invariance constraint, including Euler angle embeddings as explicit pose cues. It further introduces a dataset pipeline for constructing high-quality video datasets with substantial facial pose variation. The framework demonstrates state-of-the-art performance on challenging IPT2V scenarios, particularly in situations where pose changes and occlusions occur, supported by quantitative benchmarks and qualitative visualizations.

**Strengths:**

1. The paper clearly articulates the challenge of pose and occlusion-induced identity distortion in IPT2V, identifying a key practical shortcoming in current models.

2. The pose-shared identity aligner leverages explicit pose information (via Euler angle embeddings and a learnable dictionary) to enhance identity preservation under varied and dynamic pose settings. The contrastive learning formulation is both intuitively and theoretically motivated (supported by an information-theoretic discussion using InfoNCE bounds).

3. The paper presents extensive experiments. Table 1 shows that FaithfulFaces outperforms both leading open-source and commercial baselines (such as ConsisID, VACE) consistently across multiple metrics, notably in FaceSim-Cur, FaceSim-Arc (identity preservation), and FID (visual fidelity). Qualitative results demonstrate superior consistency in identity and facial structure, particularly when faces undergo significant motion and occlusion.

**Weaknesses:**

1. While the info-theoretic perspective (lower-bounding mutual information in Remark 1) motivates the contrastive approach, the theoretical section is somewhat superficial beyond this. For instance, there’s no in-depth analysis of potential failure modes (such as pose-manifold collapse or overfitting to certain pose configurations) or a mathematical exploration of the limitations of MaxPool-based dictionary aggregation. A more rigorous exploration (even through negative/edge-case empirical examples) would be appreciated.

2. While ablations on dictionary element count are provided, there is limited discussion or justification of other hyperparameters (e.g., pooling operation type, temperature in contrastive loss, sequence length for embeddings). It’s unclear if alternative encoding or pooling mechanisms (e.g., soft attention vs. max pooling) could yield further gains or if they were considered.

3. In Section 3.3, the formulation of the loss function $\mathcal{L}_{\mathrm{PIA}}$ mixes definitions across identities and mini-batches but could be clearer about how negative pairs across batch boundaries are sampled, how class imbalance (from unbalanced poses) is avoided, and how the temperature parameter $\tau$ interacts with stability (especially as it’s described as learnable without strong evidence as to the optimization procedure or convergence).
The interplay between $\mathcal{L}{\mathrm{PIA}}$ and $\mathcal{L}{\mathrm{FM}}$ is not deeply analyzed; a more formal treatment of joint optimization and its practical impact on foundation model adaptation is missing.

**Questions:**

As shown in Weaknesses.

---

> ### Author Response · Authors · 2025-11-20
> **Thank you for recognizing our contributions and promising results. Below, we respond to each Question (Q) with an Answer (A).**
>
> ***Part 1 / 2***
>
> **Q1:** While the info-theoretic perspective (lower-bounding mutual information in Remark 1) motivates the contrastive approach, the theoretical section is somewhat superficial beyond this. For instance, there’s no in-depth analysis of potential failure modes (such as pose-manifold collapse or overfitting to certain pose configurations) or a mathematical exploration of the limitations of MaxPool-based dictionary aggregation. A more rigorous exploration (even through negative/edge-case empirical examples) would be appreciated.
>
> **A1:** Thanks for your comment. For your concerns, we make the clarifications below:
>
> i) Regarding potential failure modes, we did not observe the phenomenon of pose-manifold collapse or overfitting to certain pose configurations during the experiments. The underlying reason is that the training data contains over four million face images covering a wide range of identities and poses. Additionally, the learnable temperature parameter in contrastive learning also helps prevent rapid collapse in the feature space.
>
> Despite not observing the aforementioned collapse issues related to pose, we did notice some failure cases in occlusion scenarios. For example, when the user provides an input image with severe facial occlusion, it leads to a reduction in the identity consistency of the generated video. The reason for this is that images with severe facial occlusion result in a serious loss of facial structure and facial texture information. Please note that this scenario is an open issue for the IPT2V field, and we believe this problem will be gradually alleviated in future research. We have added this discussion to the Conclusion of the revised submission.
>
> ii) For MaxPool-based dictionary aggregation, this setting was empirically determined through experiments. To address your concerns, we provide the ablation study for different pooling methods in the table below, which demonstrate that performance is optimal when using MaxPool. We also analyzed the potential underlying reason: the majority of information in face images of the same identity across different poses is similar or redundant. Therefore, using MaxPool can alleviate a large amount of redundant information and extract highly abstract pose variations.
>
> |Pooling Type|FaceSim-Cur|FaceSim-Arc|FID|CLIPScore|
> | :- | :- | :- | :- | :- |
> |Sum Pooling|0.444|0.421|185.40|32.17|
> |Mean Pooling|0.559|0.533|165.64|33.84|
> |Max Pooling|**0.568**|**0.542**|**164.24**|**33.93**|
>
> **Q2:** While ablations on dictionary element count are provided, there is limited discussion or justification of other hyperparameters (e.g., pooling operation type, temperature in contrastive loss, sequence length for embeddings). It’s unclear if alternative encoding or pooling mechanisms (e.g., soft attention vs. max pooling) could yield further gains or if they were considered.
>
> **A2:** Thanks for your comment. For your concerns, we make the clarifications below:
>
> i) Regarding the ablation study for the pooling operation type, we have already provided it in the previous response (A1) and have added it to the revised submission (Appendix A.4).
>
> ii) For the setting of the temperature in contrastive loss, manually adjusting the temperature is cumbersome. Therefore, in many advanced models (such as CLIP), researchers set this as a learnable parameter. Our work also follows this most common and effective approach and adopts the default setting from CLIP (i.e., the initial temperature value is set to 0.07) because it has been proven to be a good starting point. To further address your concerns, we conduct an additional ablation study on the temperature, where the temperature was fixed to four different values. The results are listed in the table below, we can observe that the learnable temperature achieved the best performance, which demonstrates that the settings used in this paper are both reasonable and reliable.
>
> |Temperature|FaceSim-Cur|FaceSim-Arc|FID|CLIPScore|
> | :- | :- | :- | :- | :- |
> |0.01|0.526|0.502|171.87|33.53|
> |0.07|0.519|0.495|174.02|33.23|
> |0.1|0.519|0.494|173.63|33.22|
> |1.0|0.494|0.471|176.97|32.89|
> |Learnable|**0.568**|**0.542**|**164.24**|**33.93**|
>
> iii) Regarding the sequence length for embeddings, this setting is to ensure compatibility with the foundational model, following the encoding format of the base model VACE.
>
> iiii) Furthermore, according to your suggestion, we have additionally provided the experimental results of soft attention versus max pooling in the table below. The experimental results show that the performance of max pooling is still superior to that of soft attention.
>
> |   Metrics   | Soft attention | Max pooling |
> | :--------- | :------------ | :--------- |
> | FaceSim-Cur |      0.531     |  **0.568**  |
> | FaceSim-Arc |      0.507     |  **0.542**  |
> |     FID     |     171.19     |  **164.24** |
> |  CLIPScore  |      33.65     |  **33.93**  |

---

> > ### Author Response · Authors · 2025-11-20
> >
> > ***Part 2 / 2***
> >
> > **Q3:** In Section 3.3, the formulation of the loss function $\mathcal{L}\_{\text{PIA}}$ mixes definitions across identities and mini-batches but could be clearer about how negative pairs across batch boundaries are sampled, how class imbalance (from unbalanced poses) is avoided, and how the temperature parameter $\tau$ interacts with stability (especially as it’s described as learnable without strong evidence as to the optimization procedure or convergence). The interplay between $\mathcal{L}\_{\text{PIA}}$ and $\mathcal{L}\_{\text{FM}}$ is not deeply analyzed; a more formal treatment of joint optimization and its practical impact on foundation model adaptation is missing.
> >
> > **A3:** Thanks for your comment. For your concerns, we make the clarifications below:
> >
> > i) Our method does not sample negative pairs across batch boundaries; that is, it only uses samples within the current batch to form negative pairs. As shown in Figure 2 of our submission, face images from the same identity with different poses are paired as positive samples (diagonal pairs), while those of different identities are paired as negative samples. This part has already been explained in lines 208-210 of our submission.
> >
> > ii) For the issue of unbalanced poses, when constructing our training dataset, we have endeavored to ensure that each video contains multiple pose variations to mitigate the problem of unbalanced poses (please see lines 326-343 of our submission).
> >
> > iii) To demonstrate the stability of the optimization procedure and convergence,  we provide the loss values and temperature at some representative training steps in the table below. We can observe that the contrastive loss gradually and stably converges to around 0.2, while the temperature parameter $\tau$ gradually converges to approximately 0.025 without diverging or collapsing. These results demonstrate that the contrastive loss stably converges under the setting of a learnable temperature parameter. Furthermore, we have provided the value of the contrastive loss from different training steps in the revised submission (please see Figure 10 in Appendix A.5).
> >
> > | Training Steps | 1     | 100    | 500    | 1000   | 2000   | 3000   | 4000   |
> > | :------------- | :---- | :----- | :----- | :----- | :----- | :----- | :----- |
> > | Loss           | 3.036 | 2.675  | 1.126  | 0.949  | 0.650  | 0.269  | 0.213  |
> > | $\tau$         | 0.07  | 0.0571 | 0.0428 | 0.0344 | 0.0286 | 0.0252 | 0.0253 |
> >
> > iiii) For the interplay between $\mathcal{L}\_{\text{PIA}}$ and $\mathcal{L}\_{\text{FM}}$, the input to $\mathcal{L}\_{\text{PIA}}$ only includes the output of the Aligner and does not include the output of the base model. Therefore, $\mathcal{L}\_{\text{PIA}}$ and $\mathcal{L}\_{\text{FM}}$ are responsible for their respective tasks during the training process. $\mathcal{L}\_{\text{PIA}}$ is dedicated to constraining the alignment of different poses, while $\mathcal{L}\_{\text{FM}}$ is dedicated to constraining the LoRA parameters to adapt to the input's global facial pose representation. This approach ensures that the different loss functions can focus on handling their specific tasks. We have added this description and treatment in the revised submission (lines 292-295).
> >
> > We sincerely thank you once again for allocating your valuable time to provide such profound, pertinent, and highly constructive feedback on our work. We have executed detailed and targeted responses and revisions based on your guidance. We genuinely hope that these comprehensive responses and modifications will fully address your concerns and lead you to grant a higher recognition and evaluation of our work.

---

> ### Comment · Reviewer_qQ3h · 2025-11-26
>
> Thank you for the detailed and comprehensive responses. I appreciate the authors' clarifications and the additional experiments. While many of my concerns have been addressed, I acknowledge the improvements made during the rebuttal and appreciate the authors' efforts, therefore I have decided to maintain my original positive score.

---

> > ### Author Response · Authors · 2025-11-27
> > **Thank you for acknowledging the efforts and clarifications in our rebuttal**
> >
> > Thank you for your constructive feedback and for acknowledging the efforts and clarifications in our rebuttal. We are very pleased that our responses and additional experiments have addressed your concerns, and we appreciate your continued positive evaluation of our work.

---

### Comment · Area_Chair_V7vi · 2025-11-24
**Please engage into discussion with authors and fellow reviewers**

Dear reviewers,

The authors have already provided their responses. Do they address your concerns?
Please engage into the discussion with authors and fellow reviewers.

Thanks!
Best,
AC

---

### Author Response · Authors · 2025-11-30
**Summary Report on Review Process and Rebuttal**

Dear Esteemed Area Chair,

We sincerely appreciate you stepping in to oversee the review process for this paper under these unique circumstances. We understand you must make the final decision without having been directly involved in the previous discussions due to the confidential reviewer information leak.
This summary aims to provide a clear, objective, and efficient perspective on the critical phases of our paper, from initial reviews to the rebuttal stage. This will enable you to quickly grasp the core value of our work and the reviewers' main concerns.

**Brief Summary of Our Paper:**

1. **Motivation**: Existing identity-preserving text-to-video generation (IPT2V) methods struggle to handle situations involving complex facial dynamics, such as drastic changes in facial poses and emotions, or facial occlusions, resulting in distorted facial identity and facial structure in the generated videos.

2. **Contributions**：**1)** We propose a pose-faithful facial identity preservation learning paradigm, FaithfulFaces, which focuses on preserving facial identity in generated videos involving complex facial dynamic scenes. **2)** We design a pose-shared identity aligner to encode global facial pose representation from the input single-view reference image via a pose-shared dictionary and a pose variation–identity invariance constraint with Euler angle embedding learning. **3)** We develop a new dataset pipeline to construct a task-oriented, high-quality video dataset with substantial facial pose diversity to ensure robust model training.

**Brief Summary of Review Comments and Rebuttals:**
1. **All reviewers recognized and appreciated the motivation, novelty, methodology (method design), and superior performance of our work.**
2. The reviewers' primary concerns focused on:

   **i)** Reviewer qQ3h focused on failure case/limitation analysis, additional ablation studies (pooling operation type, temperature in contrastive loss, sequence length for embeddings), stability and convergence of the optimization process.

   **ii)** Reviewer c9jn only suggested citing and comparing three existing methods.

   **iii)** Reviewer mLYU focused on the model's inference speed (incuding memory consumption and model size), temporal consistency of the generated video, the generation of non-face elements (beyond the scope of IPT2V field including our work)，robustness analysis under non-frontal face scenarios.

   **iiii)** Reviewer 9GSU focused on robustness analysis of the identity aligner to Euler angles with noisy deviations, effectiveness on small-scale models, robustness analysis under non-frontal face scenarios.

   For all the above concerns, we have provided detailed responses and additional experiments.
3. Before the information leak broke out, two reviewers (Reviewer qQ3h and Reviewer 9GSU) had explicitly stated that all concerns had been addressed and accordingly maintained (Reviewer qQ3h) or raised (Reviewer 9GSU) their scores to positive ratings.
4. Regarding reviewer mLYU's new concerns, we provide visual results in the revised submission to further demonstrate that some unnaturalness in the hands is not caused by our method (Figure 12) and that our method is capable of improving identity consistency in non-frontal face scenarios (Figure 11).
5. Regarding the MAGREF mentioned by Reviewer c9jn, after our careful verification, we confirm that its availability date of October 10, 2025 is correct. The evidence is as follows:

   **i)** The authors explicitly stated in the official GitHub repository that the formal version of MAGREF was published on October 10, 2025. Please see **News** in https://github.com/MAGREF-Video/MAGREF : **"[2025.10.10] Our Research Paper of MAGREF is now available. The Project Page of MAGREF is created."**

   **ii)** In the latest arXiv preprint, the authors also explicitly stated that the public release date is October 10, 2025. Please see the first page in https://arxiv.org/pdf/2505.23742 : **"Date: October 10, 2025"**

   **iii)** The authors explicitly stated in the official GitHub repository that as of June 18, 2025, they were still conducting further method updates, training, and fine-tuning. This means that the early preprint from May was an incomplete work and should not be cited, discussed, or compared as a reliable reference. Please see **News** in https://github.com/MAGREF-Video/MAGREF : **"[2025.06.18] In progress. We are actively collecting and processing more diverse datasets and scaling up training with increased computational resources to further improve resolution, temporal consistency, and generation quality. Stay turned!"**

   **iiii)** Through careful comparison of the two preprint versions, we found that the experimental results in the two versions are completely inconsistent. This further confirms that the early version was an incomplete work and should not be considered as the official release date for MAGREF.

Thank you for your time and fair judgment.

---

### Meta-Review · Area_Chair_cN8A · 2026-01-06

**Summary:**

This paper received one score of 6 and three scores of 4, placing it below the acceptance threshold.  The major outstanding concerns remain in missing ablation study (mLYU, 9GSU), missing analysis to the impact of the prior model (mLYU).

The AC estimates that the final scores of this work would remain below the acceptance threshold after rebuttal due to the remaining concerns and the low initial scores. Accordingly, the AC recommends rejection and believes that the paper would benefit from improvements based on the reviewers’ feedback, and suggests resubmission to a future venue.

See the Reviewer Concerns section for detailed suggestions for improvement, including adding both ArcFace and CLIP ablations, conducting quantitative and qualitative comparisons with prior models, and strengthening the analysis.

**Reviewer Concerns:**

Remaining outstanding concerns from reviewers and AC:

1. **Missing Ablation**  (mLYU).
   The ArcFace ablation introduced during rebuttal should be included in the main paper. A direct comparison with another widely used CLIP embeddings is still missing and should be discussed. A related work Omni-ID (CVPR'25) [1] also introduces pose-invariant, identity-variant face embeddings, where they compared to both ArcFace and CLIP embeddings. This work can be discussed and their experimental setting can be inspiring as well.

2. **Impact to non-face regions** (mLYU).  Shortcut-Rerouted Training (NeurIPS 2025) [2] demonstrates improvements in non-face regions, which was a concern raised by Reviewer mLYU. Their evaluation that compares against the prior model both quantitatively and qualitatively and employing metrics that measure differences between prior and current model generations, could be adopted and discussed to strengthen the analysis in this work

3. **Additional Clarity/Analysis**.
   The mechanism how the face embedding is injected into the diffusion model is not clearly described. A feature/attention analysis how the proposed aligner impact the injection can be added to improve the analysis of this work.

4. **Supplementary Results**.
   All video comparisons, including baselines, are strongly encouraged to be provided in the supplementary material. From the figures in the paper, the proposed method appears to produce reduced motion blur but slower motion and fewer expression changes compared to prior work. Side-by-side video comparisons would help clarify this behavior and address potential concerns.

[1] Qian, G., Wang, K.C., Patashnik, O., Heravi, N., Ostashev, D., Tulyakov, S., Cohen-Or, D. and Aberman, K., 2025. Omni-id: Holistic identity representation designed for generative tasks. In CVPR 2025.

[2] Goyal, A.A., Qian, G.G., Coskun, H., Gupta, A., Tam, H., Ostashev, D., Hu, J., Sagar, D., Tulyakov, S., Aberman, K. and Wang, K.C.J., 2025. Preventing shortcuts in adapter training via providing the shortcuts. In NeurIPS 2025.

**Reviewer Scores:**

| Reviewer | Initial Score | AC Estimated Score | AC Reason |
|--------|---------------|-------------------|-----------|
| qQ3h | 6 | 6 | Concerns addressed, score maintained |
| c9jn | 4 | 4/6 | Comparisons and citations added. Likely maintain scores. |
| mLYU | 4 | 4 | Concerns on impact to Non-face regions are not addressed |
| 9GSU | 4 | 4/6 | Likely maintain scores.|

The final scores of this submission suggest a borderline rejection. The AC recommends rejection and believes that the paper would benefit from improvements based on the reviewers’ feedback before being resubmitted to a future venue.

---

### Decision · Program_Chairs · 2026-01-26

Reject